# Accessibility of the unstructured α-tubulin C-terminal tail is controlled by microtubule lattice conformation

**Takashi Hotta[1†‡], Morgan L Pimm[1†], Ezekiel C Thomas[1†], Yang Yue[1†], Patrick DeLear[2,3], Lynne Blasius[1], Michael A Cianfrocco[4], Morgan E DeSantis[5], Ryota Horiuchi[6], Takumi Higaki[6,7], David Sept[3], Ryoma Ohi[1*‡], Kristen J Verhey[1,2]\***

[1]Department of Cell and Developmental Biology, University of Michigan, Ann Arbor, United States; [2]Department of Biophysics, University of Michigan, Ann Arbor, United States; [3]Department of Biomedical Engineering, University of Michigan, Ann Arbor, United States; [4]Department of Biological Chemistry, University of Michigan, Ann Arbor, United States; [5]Department of Molecular, Cellular and Developmental Biology, University of Michigan, Ann Arbor, United States; [6]Graduate School of Science and Technology, Kumamoto University, Kumamoto, Japan; [7]International Research Organization for Advanced Science and Technology, Kumamoto University, Kumamoto, Japan

**\*For correspondence:**
oryoma@umich.edu (RO);
kjverhey@umich.edu (KJV)

[†]These authors contributed equally to this work

**Present address:** [‡]Department of Internal Medicine, University of New Mexico Comprehensive Cancer Center, Albuquerque, United States

**Competing interest:** The authors declare that no competing interests exist.

## eLife Assessment

This **fundamental** work reveals that the accessibility of the unstructured C-terminal tail of α-tubulin differs with the state of the microtubule lattice. Accessibility increases with the expansion of the lattice induced by GTP and certain MAPs, which can then dictate the subsequent interactions between MAPs and microtubules, and post-translational modifications of tubulin tails. The evidence supporting the conclusion is **compelling**, although the characterisation of the probes does not answer whether they directly affect the lattice or expose the C-terminal tail of α-tubulin. The probes can be used as tools in the future to study differences in microtubule lattice regulation under different conditions both in vitro and in vivo. This work will be of great interest to the cytoskeleton field.

**Abstract** Microtubules are cytoskeletal filaments that self-assemble from the protein tubulin, a heterodimer of α-tubulin and β-tubulin, and are important for cell mechanics, migration, and division. Much work has focused on how the nucleotide state of β-tubulin regulates the structure and dynamics of microtubules. In contrast, less is known about the structure and function of the C-terminal tails (CTTs) of α- and β-tubulin which are thought to freely protrude from the surface of the microtubule. To study the CTT of α-tubulin, we developed three different biosensors that bind the tyrosinated α-tubulin CTT (Y-αCTT). Surprisingly, live imaging of the probes indicates that the Y-αCTT is minimally accessible along the microtubule lattice under normal cellular conditions. Lattice binding of the Y-αCTT probes can be increased by three different ways of changing the tubulin conformational state: the drug Taxol, expression of microtubule-associated proteins (MAPs) that recognize or promote an expanded tubulin conformation, or expression of tubulin that cannot hydrolyze GTP. Molecular dynamics simulations indicate that the Y-αCTT undergoes numerous transient interactions with the bodies of α-tubulin and β-tubulin in the lattice, and that the frequency of these interactions is regulated by the tubulin nucleotide state. These findings suggest that accessibility of the Y-αCTT is locally governed by nucleotide- and MAP-dependent conformational changes to tubulin subunits within the microtubule lattice.

## Introduction

Microtubules are dynamic polymers that provide mechanical support to cells, serve as tracks for intracellular trafficking, and form the mitotic spindle that separates the replicated genome during cell division (*Forth and Kapoor, 2017*; *Fourriere et al., 2020*; *Logan and Menko, 2019*; *Prosser and Pelletier, 2017*; *Risteski et al., 2021*). Microtubules are formed by self-assembly of the protein tubulin, a heterodimer of α-tubulin and β-tubulin. Although both α- and β-tubulin bind GTP and thus belong to the superfamily of G proteins (*Hughes, 1983*; *Nogales et al., 1998*), only β-tubulin hydrolyzes GTP and exchanges GDP for GTP.

The ability of β-tubulin to hydrolyze GTP is intimately linked to microtubule assembly and disassembly (*Chew and Cross, 2025*; *Cleary and Hancock, 2021*; *Gudimchuk and McIntosh, 2021*). During polymerization, tubulin subunits with GTP bound to β-tubulin (GTP-tubulin) associate in a head-to-tail fashion with the plus (growing) end of the microtubule. After incorporation into the microtubule lattice, β-tubulin hydrolyzes its GTP to GDP. When GDP-tubulin subunits are exposed at the plus end, microtubules are unstable and depolymerization ensues. Thus, a cap of GTP-tubulin stabilizes the microtubule lattice (*Mitchison and Kirschner, 1984*; *Figure 1A and B*).

Recent work suggests that GTP hydrolysis by β-tubulin triggers a decrease in the spacing of tubulin subunits along the longitudinal axis of the microtubule, largely due to conformational changes in the associated α-tubulin subunit. Specifically, cryo-electron microscopy (cryo-EM) of microtubules assembled in vitro shows GTP- or GTP-like tubulin to be in an expanded state (dimer length 83–84 Å), whereas GDP-Pi or GDP-tubulin is in a compacted state (dimer length 81–82 Å; *Alushin et al., 2014*; *Estevez-Gallego et al., 2025*; *Estévez-Gallego et al., 2020*; *LaFrance et al., 2022*; *Manka and Moores, 2018*; *Zhang et al., 2018*; *Figure 1B*). In cells, most microtubules have a compacted lattice (dimer length ~82 Å), consistent with their being in the GDP state (*de Jager et al., 2025*).

GDP-tubulin subunits within the lattice can be switched from their compacted state to a GTP-like expanded state by tubulin-binding drugs and microtubule-associated proteins (MAPs). Taxol is a natural product that stabilizes microtubules and cryo-EM studies showed that Taxol expands the microtubule lattice in vitro (dimer length ~84 Å) and in cells (dimer length ~85 Å; *Alushin et al., 2014*; *de Jager et al., 2025*; *Kellogg et al., 2017*; *Prota et al., 2023*). MAP binding can also alter tubulin conformational state, although in these studies the lattice state has largely been deduced from light microscopy experiments and thus the exact changes in dimer length are unclear. The MAPs kinesin-1 and CAMSAP3 recognize and/or induce an expanded microtubule lattice (*de Jager et al., 2025*; *Liu and Shima, 2023*; *Peet et al., 2018*; *Shen and Ori-McKenney, 2024*; *Shima et al., 2018*), whereas the MAPs tau, MAP2, and doublecortin (DCX) recognize and/or induce a compacted microtubule lattice (*Castle et al., 2020*; *Paquette et al., 2025*; *Siahaan et al., 2022*). In cells, microtubule lattice conformation can also be altered upon changes in the osmotic environment (*Shen and Ori-McKenney, 2024*). At least three outcomes have been linked to conformational plasticity of tubulin subunits within the lattice (*Chew and Cross, 2025*; *Verhey and Ohi, 2023*). First, tubulin isotypes may differ in their ability to undergo compaction and expansion and thereby impact cell- and tissue-specific responses to drugs and MAPs (*Chew and Cross, 2023*; *Howes et al., 2017*). Second, molecular motor proteins that step along the microtubule surface can communicate with each other through the lattice (*Muto et al., 2005*; *Wijeratne et al., 2022*). Third, microtubule lattice state regulates the tubulin code by permitting specific subsets of microtubules to be marked by tubulin post-translational modifications (PTMs; *Egoldt et al., 2025*; *Shen and Ori-McKenney, 2024*; *Yue et al., 2023*; *Yue et al., 2025*).

A critical element of both α- and β-tubulins is their C-terminal tails (CTTs) (*Figure 1A*), disordered and negatively charged segments that extend from the body of the tubulin proteins. The CTTs alter microtubule dynamics by impeding microtubule polymerization and increasing the growth-to-shortening transition (*Serrano et al., 1984b*). The CTTs are thought to be a major control hub of the microtubule lattice as they contain the majority of sequence differences between tubulin isotypes and contain over 40% of the overall charge of tubulin. They are also hotspots for PTMs that are broadly important for cellular processes including intracellular trafficking, cell division, ciliary beating, neuronal pathfinding, and cardiomyocyte function (*Janke and Magiera, 2020*; *Roll-Mecak, 2020*). In addition, the CTTs regulate binding and activity of MAPs, motor proteins, and severing proteins (*Fan and McKenney, 2023*; *Lindsay et al., 2023*; *McKenney et al., 2016*; *Niederstrasser et al., 2002*; *Serrano et al., 1984a*; *Serrano et al., 1985*; *Wang and Sheetz, 2000*; *Zehr et al., 2020*). Despite their importance for microtubule biochemistry and function, little is known about the structure or

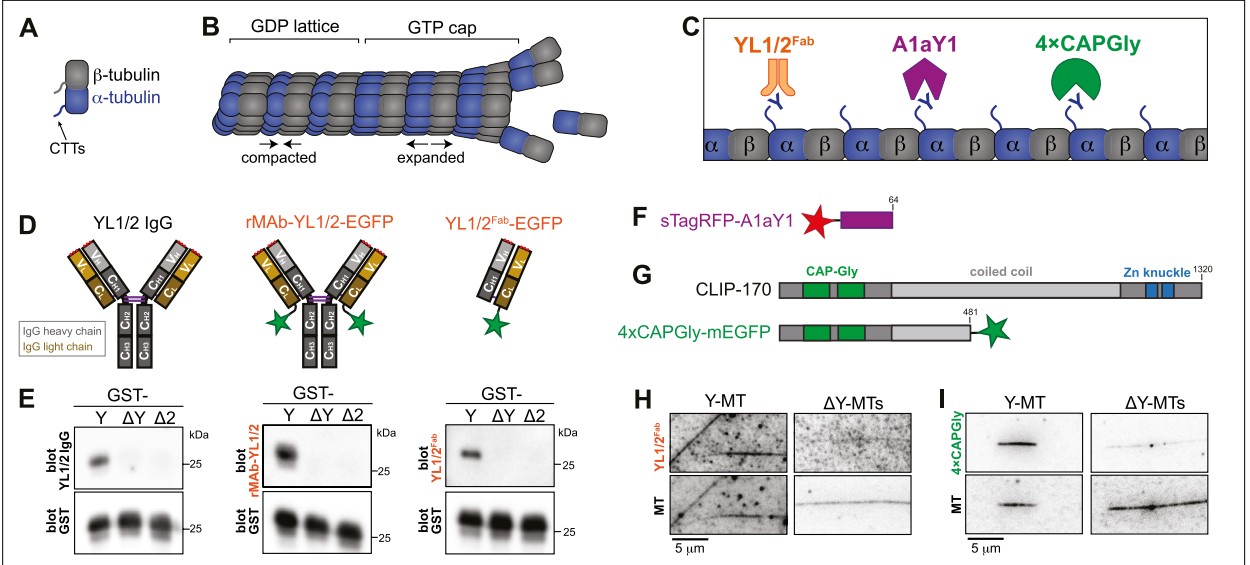

**Figure 1.** Three probes that recognize the Y-αCTT. (**A, B**) Schematic of tubulin protein and its conformational states within the microtubule lattice. (**A**) Schematic of tubulin heterodimer with the unstructured C-terminal tails (CTTs) protruding from the body of α- and β-tubulin. (**B**) Tubulin adds to the end of a microtubule in a GTP-bound and expanded state, resulting in a stabilizing GTP cap. In the microtubule lattice, β-tubulin undergoes GTP hydrolysis, resulting in a GDP lattice and compaction of α-tubulin. (**C**) Schematic of three sensors (YL1/2, A1aY1, and 4xCAPGly) generated to detect the accessibility of the Y-αCTT along the microtubule lattice. (**D, E**) Generation and validation of the YL1/2^Fab probe. (**D**) Schematic of antibody proteins. YL1/2 IgG: typical mammalian IgG molecule containing two heavy (H) and two light (L) chains. Light chains are comprised of one variable (V_L, light orange) and one constant (C_L, dark orange) region. Heavy chains are comprised of one variable (V_H, light gray) and three constant (C_{H1-3}, dark gray) regions. Red dots indicate complementarity determining regions (CDRs) and purple lines indicate disulfide bonds. rMAb-EGFP: recombinant monoclonal antibody (rMAb) with EGFP (green star) fused to the C-terminus of the light chain. YL1/2^Fab-EGFP: Fragment antibody binding (Fab) produced from rMAb-EGFP by papain cleavage. (**E**) GST-tagged αCTT sequences were probed by western blotting with (left) commercial YL1/2 monoclonal antibody, (middle) rMAb-YL1/2-EGFP, or (right) YL1/2^Fab-EGFP. Each blot was also probed for GST protein as a loading control. Y: full-length and tyrosinated αCTT sequence; ΔY: detyrosinated αCTT sequence (lacking the C-terminal tyrosine); ΔC2: αCTT sequence lacking the C-terminal two amino acids. (**F**) Schematic of synthetic protein A1aY1 tagged at its N-terminus with sTag-RFP (red star). (**G**) Schematic of the domain organization of (top) full-length CLIP-170 and (bottom) the 4xCAPGly probe tagged at its C-terminus with mEGFP (green star). (**H, I**) Representative images of (**H**) YL1/2^Fab-EGFP or (**I**) 4xCAPGly-mEGFP proteins binding to tyrosinated (Y–MT) or detyrosinated (ΔY-MT) microtubules. HeLa tubulin was polymerized into microtubules and Taxol-stabilized. The microtubules were used directly (Y-MTs) or detyrosinated by VASH/SVBP-containing lysate before adding the (**H**) YL1/2^Fab-EGFP or (**I**) 4xCAPGly-mEGFP probes. Scale bars: 5 μm.

The online version of this article includes the following source data and figure supplement(s) for figure 1:

**Source data 1.** TIFF files of original western blots.

**Source data 2.** PDF file containing original gels for panel E indicating the relevant bands.

**Figure supplement 1.** Recombinant YL1/2 antibody and purified probes.

**Figure supplement 1—source data 1.** PDF file containing original gels for panels B and C indicating the relevant bands.

**Figure supplement 1—source data 2.** TIFF files of original western blots.

**Figure supplement 2.** The A1aY1 probe must be imaged in live cells.

**Figure supplement 3.** The 4xCAPGly probe must be imaged in live cells.

positioning of the CTTs of α- and β-tubulin. Indeed, the CTTs are assumed to freely extend from the microtubule surface.

Interestingly, molecular modeling approaches suggest that the CTTs undergo interactions with the body of tubulin within the lattice (*Freedman et al., 2011*; *Laurin et al., 2017*; *Luchko et al., 2008*; *Sahoo and Hanson, 2025*) and thus may not always be exposed along the microtubule surface. To examine the accessibility of the αCTT along the microtubule lattice, we developed a set of three structurally unrelated biosensors that recognize the native (unmodified) CTT sequence of α-tubulin isotypes that contain a C-terminal tyrosine residue (tyrosinated or Y-αCTT; *Figure 1C*). The first probe is a fragment antigen binding (Fab) of the rat monoclonal antibody YL1/2 (*Kilmartin et al., 1982*; *Wehland et al., 1983*). The second probe is A1aY1, a synthetic protein isolated using yeast display

to identify proteins that recognize the CTT sequence of the α-tubulin isotype TubA1A (*Kesarwani et al., 2020*). The third probe contains the first two CAP-Gly domains of the cytoplasmic linker protein CLIP-170 and was designed based on the known ability of CAP-Gly domains to bind to the Y-αCTT (*Bieling et al., 2008*; *Honnappa et al., 2006*; *Mishima et al., 2007*; *Peris et al., 2006*; *Weisbrich et al., 2007*). Notably, the three probes are derived from different protein sources, lending rigor and robustness to our results.

Strikingly, we find that the Y-αCTT probes show limited binding to microtubules when imaged in live cells, suggesting that the α-tubulin CTT is minimally accessible along most microtubules in cells. We find that binding of the Y-αCTT probes to microtubules is increased upon fixation, taxol treatment, or binding of MAPs that induce an expanded lattice state. We also find that microtubule binding of the Y-αCTT probes is enhanced by GTP-locked tubulins in cells but not in vitro. Molecular dynamics simulations suggest that the Y-αCTT undergoes more interactions with the tubulin body when tubulin is in the compacted state. These findings suggest that accessibility of the αCTT can be locally regulated by changes to the tubulin conformational state.

## Results

### Biosensors of αCTT accessibility in the microtubule lattice

We first tested whether the rat monoclonal antibody YL1/2 (*Kilmartin et al., 1982*; *Wehland et al., 1983*) could serve as a sensor of Y-αCTT availability. We determined the amino acid sequence of the immunoglobulin G (IgG) heavy and light chains of YL1/2 (*Figure 1D*, *Figure 1—figure supplement 1A*) and then purified an EGFP-tagged recombinant monoclonal antibody (rMAb) version of YL1/2 (rMAb-YL1/2-EGFP, *Figure 1D*, *Figure 1—figure supplement 1B*). To confirm that the experimentally determined protein sequences generate recombinant protein with efficacy and specificity identical to the original antibody, we tested the ability of rMAb-YL1/2-EGFP to recognize the proper antigen using western blotting. We purified glutathione-S-transferase (GST) proteins fused with different human TubA1A αCTT sequences, either full-length [tyrosinated (Y): SVEGEGEEEGEEY], lacking the C-terminal tyrosine [detyrosinated (ΔY): SVEGEGEEEGEE] or lacking the terminal two amino acids (Δ2: SVEGEGEEEGE). The rMAb-YL1/2-EGFP protein recognized the GST-Y sequence by western blot, but not the GST-ΔY or GST-ΔC2 sequences, identical to the commercial YL1/2 antibody (*Figure 1E*). To avoid potential issues with bivalency, we then generated a monovalent EGFP-tagged Fragment antibody binding (Fab) version (*Figure 1D*) by papain digestion and confirmed that the YL1/2[Fab]-EGFP protein retains the ability to specifically recognize the GST-Y sequence by western blotting (*Figure 1E*).

We next tested A1aY1, a synthetic protein isolated using yeast display to identify proteins that recognize the TubA1A CTT (*Kesarwani et al., 2020*). We used an A1aY1 probe tagged with super-TagRFP (sTagRFP) (*Figure 1F*) as a similar construct with TagRFP-T was previously shown to provide strong recognition of the Y-αCTT (*Kesarwani et al., 2020*). We were unable to generate recombinant sTagRFP-A1aY1 probe as the protein was prone to aggregation, perhaps due to the oligomeric nature of the sTagRFP moiety (*Mo et al., 2020*). We thus tested A1aY1 by transient expression in COS-7 cells whose flat morphology enhances visualization of microtubules. We found that sTagRFP-A1aY1 shows little to no microtubule binding in cells that are imaged live (*Figure 1—figure supplement 2A*) but localizes more strongly along microtubules in fixed cells (*Figure 1—figure supplement 2B and C*). We also found that probe localization was sensitive to expression levels as the sTagRFP-A1aY1 probe was diffusely localized at low levels of expression but could decorate the microtubule lattice at higher levels of expression (not shown). To minimize variability in expression levels, we generated a HeLa-Kyoto stable cell line that expresses sTagRFP-A1aY1 in a doxycycline-inducible manner.

We then tested whether a CAP-Gly domain could act as a sensor of Y-αCTT accessibility based on the known ability of CAP-Gly domains to bind to the Y-αCTT (*Bieling et al., 2008*; *Honnappa et al., 2006*; *Mishima et al., 2007*; *Peris et al., 2006*; *Weisbrich et al., 2007*). We generated a construct containing the two tandem CAP-Gly domains and part of the first coiled-coil segment of rat CLIP170 (amino acids 3–484) tagged with mScarlet3 (mSc3; *Figure 1G*) based on a previous 'head' domain fragment called H2 (*Arnal et al., 2004*; *Bieling et al., 2008*; *Scheel et al., 1999*). The presence of the coiled-coil segment results in a dimeric protein that contains four CAP-Gly domains (hereafter referred to as 4xCAPGly). We purified recombinant 4xCAPGly-mEGFP protein from *E. coli* (*Figure 1—figure supplement 1B*) and used a far-western blot to verify its specificity for the tyrosinated αCTT sequence

of the GST-Y construct (*Figure 1—figure supplement 1C*). When transiently expressed in COS-7 cells and imaged live, 4xCAPGly-mSc3 localized to microtubule plus ends (*Figure 1—figure supplement 3A*) due to the presence of a SxIP motif which binds to end binding (EB) proteins located at the growing microtubule plus ends (*Bieling et al., 2008*). Like A1aY1, fixation caused 4xCAPGly-mSc3 to localize along microtubules (*Figure 1—figure supplement 3B and C*). We found that 4xCAPGly-mSc3 localization was also sensitive to expression levels as the probe was diffusely localized at low levels of expression but could decorate the microtubule lattice at higher levels of expression (not shown). To minimize variability in expression levels, we generated HeLa-Kyoto stable cell lines that express 4xCAPGly tagged with either mSc3 or mEGFP in a doxycycline-inducible manner.

Finally, we tested the specificity of the Y-αCTT probes in reconstitution assays. For these experiments, we purified tubulin from HeLa S3 cells which are largely devoid of tubulin PTMs including detyrosination of the αCTT (*Souphron et al., 2019*; *Thomas et al., 2025*). We assembled microtubules from HeLa tubulin and used Taxol to stabilize them in a flow chamber. The Taxol-MTs were either untreated to maintain the tyrosinated state (Y-MTs) or were treated with VASH1/SVBP to generate detyrosinated microtubules (ΔY-MTs; *Thomas et al., 2025*). Both the YL1/2^Fab-EGFP and 4xCAPGly-mSc3 probes showed stronger decoration of Y-MTs than ΔY-MTs (*Figure 1H and I*). Taken together, these results demonstrate that we have identified three different probes that specifically recognize the native and unmodified (tyrosinated) α-tubulin CTT.

## The αCTT is minimally accessible within a GDP microtubule lattice

Our initial finding that localization of the A1aY1 and 4xCAPGly probes shifts from cytosolic (A1aY1) or microtubule plus end (4xCAPGly) to the microtubule lattice upon fixation (*Figure 1—figure supplements 2 and 3*) suggests that lattice state impacts the accessibility of the Y-αCTT. Previous work showed that localization of A1aY1 shifts from cytosolic or plus ends to the microtubule lattice in cultured cells treated with SiR-Tubulin (*Kesarwani et al., 2020*). As SiR-tubulin is a Taxol derivative, it may also convert GDP-tubulin into a GTP-like expanded state (*Siahaan et al., 2022*). We thus hypothesized that lattice expansion increases the accessibility of the Y-αCTT along the GDP microtubule lattice.

To test this idea, we used live-cell imaging to examine the localization of the Y-αCTT probes before and after Taxol treatment. Given the requirements of live-cell imaging and low probe expression, we utilized the stable HeLa-Kyoto cell lines to inducibly express low levels of the probes sTagRFP-A1aY1 (A1aY1 hereafter for brevity) or 4xCAPGly-mEGFP (4xCAPGly hereafter for brevity). In untreated cells, the A1aY1 probe localized diffusely throughout the cell with faint staining along the microtubule lattice observed in some cells (*Figure 2A*, before). No change was observed upon addition of DMSO vehicle control (*Figure 2A and B*). Strikingly, within minutes of Taxol addition, localization of the A1aY1 probe to the microtubule lattice increased 1.6-fold (*Figure 2A, C and D*), suggesting that the Y-αCTT is more accessible to the probe after Taxol-induced expansion of the microtubule lattice. To verify these results in another cell line, we transiently expressed the A1aY1 probe in HeLa and COS-7 cells and found a similar increase in microtubule lattice decoration after Taxol treatment (*Figure 2—figure supplement 1A*).

We carried out similar experiments to examine the response of the 4xCAPGly probe to Taxol treatment. In untreated cells, the 4xCAPGly probe localized to the growing plus ends of microtubules via its association with endogenous EB proteins (*Figure 2E*, before), consistent with previous work (*Bieling et al., 2008*). No change was observed upon addition of DMSO vehicle control (*Figure 2E and F*). However, within minutes of Taxol treatment, the 4xCAPGly probe relocalized from the ends to the lattice of the microtubules (*Figure 2E and G*), resulting in a 1.5-fold increase in 4xCAPGly probe intensity along the lattice (*Figure 2H*). Relocalization to the microtubule lattice was specific to 4xCAPGly as EB3 protein was lost from the microtubule ends and did not accumulate along the microtubule lattice upon Taxol treatment (*Figure 2—figure supplement 1C*). These results suggest that Taxol-mediated expansion of the microtubule lattice renders the Y-αCTT accessible for 4xCAPGly probe binding. Similar results were obtained when the 4xCAPGly probe was transiently expressed in HeLa or COS-7 cells (*Figure 2—figure supplement 1B*).

Taken together, these results suggest that the Y-αCTT is minimally accessible when tubulin in the microtubule lattice is in the GDP and compacted state but is more accessible for probe binding when GDP-tubulin subunits are pharmacologically converted into a GTP-like expanded state.

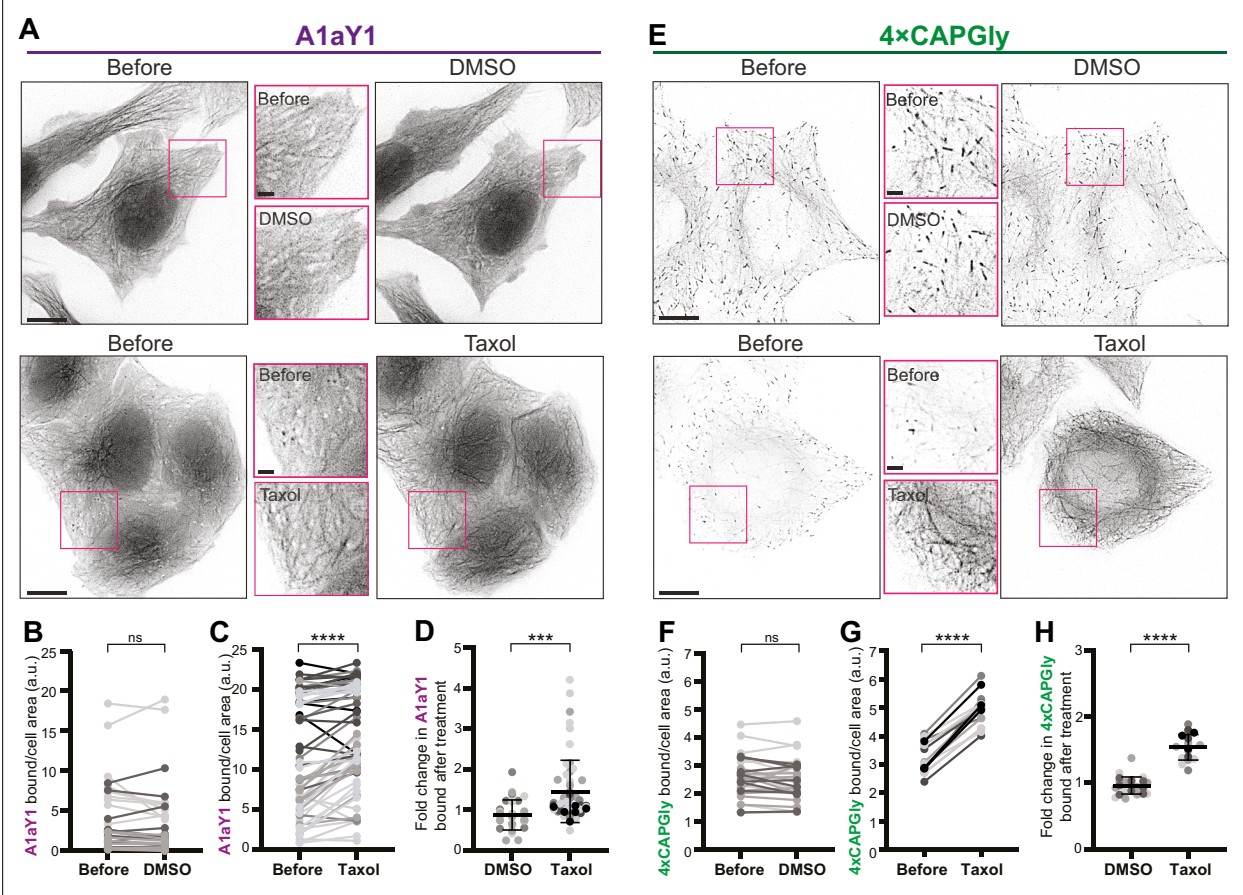

**Figure 2.** Y-αCTT probes bind to the microtubule lattice after Taxol treatment. (**A–D**) Live-cell imaging of A1aY1 probe. (**A**) Representative images of the sTagRFP-A1aY1 HeLa stable cell line before and 15 min after treatment with (top) 0.3% DMSO control or (bottom) 10 μM Taxol. Insets (boxes) show magnified views of A1aY1 probe. Scale bars: 10 μm in whole-cell views and 2 μm in magnified views. (**B, C**) Quantification of probe binding to microtubules. Paired data plots display the amount of A1aY1 probe bound to microtubules in individual cells before and after treatment with (**B**) DMSO or (**C**) Taxol. (**D**) Mean difference plot showing the fold change in A1aY1 probe binding in DMSO- vs Taxol-treated cells. DMSO = 29 cells across three experiments; Taxol = 59 cells across seven experiments. (**E–H**) Live-cell imaging of 4xCAPGly probe. (**E**) Representative images of the 4xCAPGly-mEGFP HeLa stable cell line before and 15 min after treatment with (top) 0.3% DMSO control or (bottom) 10 μM Taxol. Insets (boxes) show magnified views of CAPGly probe. Scale bars: 10 μm in whole-cell views and 2 μm in magnified views. (**F, G**) Quantification of probe binding to microtubules. Paired data plots display the amount of 4xCAPGly probe bound to microtubules in individual cells before and after treatment with (**F**) DMSO or (**G**) Taxol. (**H**) Mean difference plot showing the fold change in 4xCAPGly probe binding in DMSO- or Taxol-treated cells. DMSO = 28 cells across three experiments; Taxol = 17 cells across five experiments. Error bars in (**D**) and (**H**) indicate SD. ***: $p<0.0002$; ****: $p<0.0001$; ns: not significant Student's t test (**B, C, F, G**: two-tailed; paired), (**D, H**: unpaired).

The online version of this article includes the following source data and figure supplement(s) for figure 2:

**Source data 1.** Excel file with fluorescence intensity measurements.

**Figure supplement 1.** Controls for probe binding in response to Taxol-mediated lattice expansion.

## MAPs that locally expand the lattice result in a local increase in αCTT accessibility and detyrosination

In cells, GDP-tubulin can be locally converted into a GTP-like expanded state by binding of specific MAPs to the microtubule lattice. We thus tested whether MAP binding can locally change the ability of the Y-αCTT probes to recognize the microtubule lattice. We transfected the stable A1aY1 and 4xCAPGly cell lines with MAPs known to recognize and/or induce an expanded state. We utilized a rigor version of kinesin-1 (KIF5C[rigor], *de Jager et al., 2025*; *Peet et al., 2018*; *Shima et al., 2018*) as well as CAMSAP2 and CAMSAP3 (*Liu and Shima, 2023*; *Yue et al., 2023*). We also included MAP7 as this MAP was recently suggested to induce and/or preserve an expanded lattice state based on its ability to increase lattice binding of the taxane SiR-tubulin upon overexpression in BEAS-2B cells

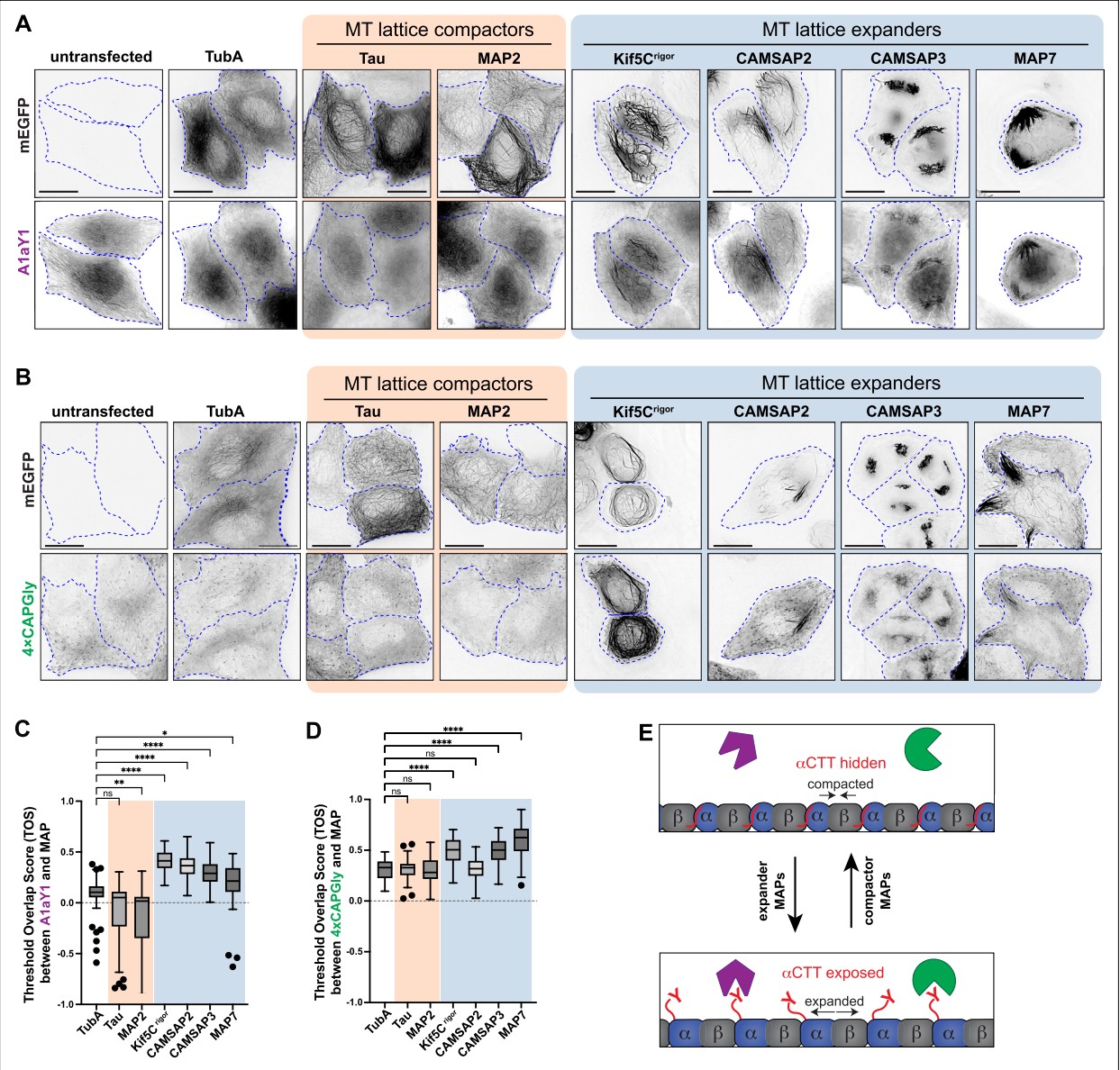

**Figure 3.** MAPs that expand the microtubule lattice increase Y-αCTT probe binding. (**A, B**) Representative live-cell images of (**A**) A1aY1 or (**B**) 4xCAPGly HeLa stable cell lines transiently expressing the indicated mEGFP-tagged tubulin or MAP constructs. Cell boundaries are indicated by blue dotted lines. Scale bars: 20 μm. (**C, D**) Quantification of (**C**) sTag-RFP-A1aY1 or (**D**) CAPGly-mSc3 probe colocalization with mEGFP-tagged tubulin or MAP constructs. The threshold overlap score (TOS) was measured on a per-cell basis where 1.0 indicates perfect colocalization, –1.0 indicates perfect anti-colocalization, and values near 0 indicate no relationship. Data from three independent experiments are presented as Tukey box plots. The box encompasses the 25th to 75th percentiles, with a line at the median. Whiskers show the last data point within 1.5 times the interquartile range. Outliers are plotted as individual points. *: $p<0.1$; **: $p<0.001$; ****: $p<0.0001$; ns: not significant (Kruskal-Wallis test followed by post-hoc Dunn's multiple pairwise comparisons with TubA1A as the control). Number of cells analyzed (n) in (**C**): TubA1A=69, Tau = 57, MAP2=66, Kif5C$^{rigor}$ = 70, CAMSAP2=50, CAMSAP3=56, and MAP7=55 and in (**D**): TubA1A=62, Tau = 61, MAP2=61, Kif5C rigor = 76, CAMSAP2=62, CAMSAP3=71, and MAP7=57. (**E**) Schematic model depicting how expander and compactor MAPs regulate microtubule lattice conformation, influencing Y-αCTT accessibility.

The online version of this article includes the following source data for figure 3:

**Source data 1.** Excel file with fluorescence intensity measurements.

(***Shen and Ori-McKenney, 2024***). As controls, we expressed the MAPs tau and MAP2 that recognize and/or induce a compacted lattice state (***Castle et al., 2020***; ***Siahaan et al., 2022***).

Expression of MAPs that expand the microtubule lattice (KIF5C$^{rigor}$, CAMSAP2, CAMSAP3, or MAP7) resulted in colocalization of the A1aY1 probe with the expanded lattice (***Figure 3A and C***), whereas expression of MAPs that compact the microtubule lattice did not change (tau) or decreased

(MAP2) localization of the A1aY1 probe along microtubules as compared to the control conditions (untransfected or mEGFP-tagged human TubA1A expression; *Figure 3A and C*). Similar results were found with the 4xCAPGly probe: only expression of the expander MAPs KIF5C$^{rigor}$, CAMSAP3, or MAP7 resulted in a significant increase in 4xCAPGly probe binding along the microtubule lattice (*Figure 3B and D*). These results indicate that in cells, MAPs that expand the lattice expose the α-tubulin CTT, whereas MAPs that compact the lattice do not. We suggest that MAP binding drives a local expansion of the underlying microtubule lattice that results in local switching of the αCTT into an accessible state recognized by the Y-αCTT probes (*Figure 3E*).

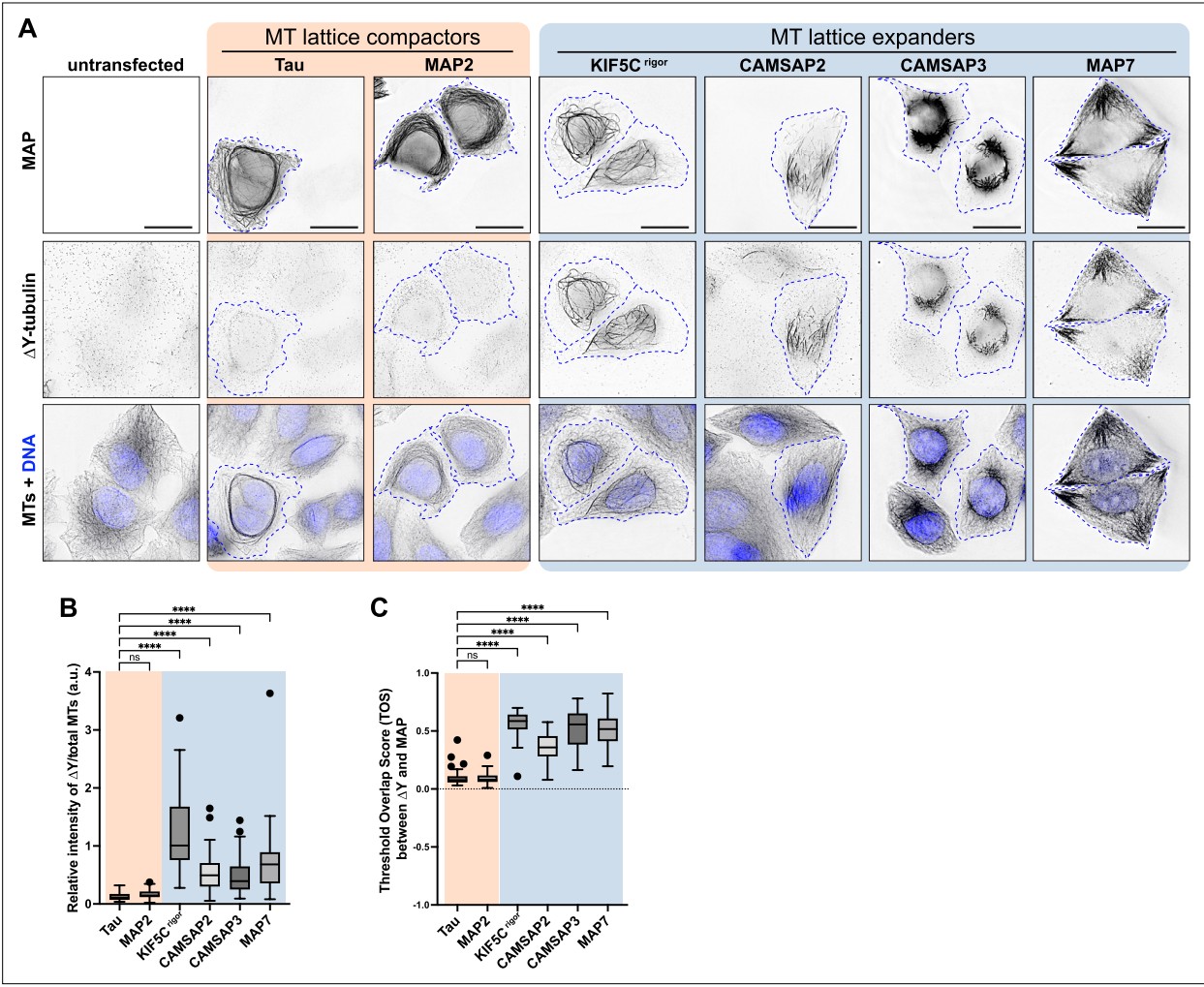

**Figure 4.** MAPs that expand the MT lattice increase detyrosination of the Y-αCTT. (**A**) Representative images of HeLa cells transiently expressing the indicated mEGFP-tagged MAPs and then fixed and stained with antibodies against detyrosinated microtubules (ΔY-tubulin) and total microtubules (MTs). Images are shown in inverted grayscale. The nuclei are represented by blue pseudocolor in the bottom panels. Blue dotted lines: boundaries of cells expressing the corresponding MAPs. Scale bars: 20 μm.(**B**) Quantification of the intensity of detyrosination on MAP-bound microtubules. The fluorescence intensity of detyrosination was measured on MAP-decorated microtubules and normalized against the total microtubule intensity of MAP-decorated microtubules. Data from three independent experiments are presented as Tukey box plots. ****: p<0.0001; ns: not significant (Kruskal-Wallis test followed by post-hoc Dunn's multiple pairwise comparisons with tau as the control). Number of cells analyzed (n): Tau = 67, MAP2=70, Kif5Cr$^{igor}$ = 75, CAMSAP2=67, CAMSAP3=70, and MAP7=67. (**C**) Quantification of the colocalization of MAPs and detyrosinated microtubules. The threshold overlap score (TOS) was measured on a per-cell basis. Data from three independent experiments are presented as Tukey box plots. The box encompasses the 25th to 75th percentiles, with a line at the median. Whiskers show the last data point within 1.5 times the interquartile range. Outliers are plotted as individual points. ****: p<0.0001; ns: not significant (Kruskal-Wallis test followed by post-hoc Dunn's multiple pairwise comparisons with tau as the control). Number of cells analyzed (n): Tau = 70, MAP2=72, Kif5Cr$^{igor}$ = 76, CAMSAP2=66, CAMSAP3=69, and MAP7=67.

The online version of this article includes the following source data for figure 4:

**Source data 1.** Excel file with fluorescence intensity measurements.

The ability of MAPs to locally regulate Y-αCTT accessibility suggests that they could regulate downstream events such as posttranslational modification of the αCTT. Consistent with this possibility, the expander MAPs CAMSAP2 and CAMSAP3 enable detyrosination (ΔY) of α-tubulin subunits by the enzyme VASH1/SVBP (*Yue et al., 2023*). To extend these findings, we tested whether the same expander and compactor MAPs that regulate Y-αCTT accessibility (*Figure 3*) cause an increase in ΔY-MTs. Indeed, like CAMSAP2 and CAMPSAP3 (*Yue et al., 2023*), expression of the expander MAPs KIF5C[rigor] or MAP7 resulted in significant increase in ΔY-MTs (*Figure 4A and B*) with the increase in ΔY-MTs localized along the MAP-bound microtubules (*Figure 4A and C*). In contrast, expression of the compactor MAPs tau or MAP2 did not cause an increase in ΔY-MTs (*Figure 4A and B*) even though, in the case of tau, overexpression caused the microtubules to be bundled in the perinuclear region of the cell (*Figure 4A*).

## Nucleotide state regulates αCTT accessibility and detyrosination

We next tested whether the GTP- vs GDP-state of tubulin can regulate accessibility of the αCTT. To test this in the stable A1aY1 and 4xCAPGly cell lines, we generated a plasmid that expresses PA-tagged human TubA1A with an internal ribosome entry site (IRES) driving the expression of an EGFP reporter protein. We compared WT TubA1A to the mutant E254A which prevents GTP hydrolysis by β-tubulin and therefore locks tubulin in a GTP-bound state (*Beckett and Voth, 2023*; *LaFrance et al., 2022*; *Roostalu et al., 2020*).

We transfected the stable A1aY1 and 4xCAPGly cell lines with plasmids for expressing TubA1A(WT) or TubA1A(E254A) and examined the localization of the Y-αCTT probes by live-cell microscopy. Western blotting showed that the tubulin proteins were expressed at comparable levels (*Figure 5— figure supplement 1*). In cells expressing WT TubA1A (visualized by expression of the EGFP reporter), the A1aY1 probe localized in a diffuse manner, whereas in cells expressing TubA1A(E254A), the A1aY1 probe localized along the microtubule lattice (*Figure 5A*). Similar results were found for the 4xCAPGly probe which localized to microtubule plus ends in cells expressing TubA1A(WT) but localized along the lattice in cells expressing TubA1A(E254A) (*Figure 5C*). Quantification of these results confirmed that expression of GTP-locked E254A α-tubulin resulted in a significant increase in the density of both A1aY1 and 4xCAP-Gly probes along the microtubule lattice (*Figure 5B and D*).

We also examined whether GTP-locked tubulin could cause an increase in detyrosination of the exposed Y-αCTT. We found that expression of GTP-locked E254A α-tubulin but not WT α-tubulin resulted in an increase in ΔY-tubulin levels (*Figure 5E and F*). These results suggest that the nucleotide state of tubulin regulates Y-αCTT accessibility for probe binding and PTM enzymes.

## MAPs but not nucleotide state regulate αCTT accessibility in vitro

We then used reconstitution experiments to directly probe the link between nucleotide state and accessibility of the αCTT. We polymerized HeLa S3 tubulin in the presence of the GTP analog guanosine-5'-[(α,β)-methyleno]tri-phosphate (GMPCPP) to generate microtubules in the GTP-like expanded state (GMPCPP-MTs) or in the presence of GTP which is hydrolyzed to generate the GDP compacted state (GDP-MTs). Furthermore, to directly compare Y-αCTT probe binding between GMPCPP-MTs and GDP-MTs in the same flow chamber, we labeled GMPCPP-MTs with fluorescent Alexa Fluor 568-labeled tubulin and GDP-MTs with fluorescent Alexa Fluor 647-labeled tubulin. We mixed the two microtubule populations with purified Y-αCTT probe proteins (*Figure 1—figure supplement 1B*) in the same chamber and stabilized the microtubules against depolymerization with 25% glycerol. We found that the purified 4xCAPGly probe bound to both microtubule populations (*Figure 6A*) with no significant difference in steady-state binding (*Figure 6B*). Similar results were obtained for the YL1/2[Fab] protein which bound to both GMPCPP- and GDP-MTs in the flow chamber (*Figure 6C*) with no significant difference in steady-state binding (*Figure 6D*). Thus, although nucleotide state can directly influence tubulin conformational state in cryoEM structures of in vitro polymerized microtubules (*Alushin et al., 2014*; *LaFrance et al., 2022*; *Manka and Moores, 2018*; *Zhang et al., 2018*), it does not appear to directly regulate accessibility of the α-tubulin CTT within the microtubule lattice.

The inability of the in vitro polymerized microtubules (*Figure 6A–D*) to recapitulate the influence of nucleotide state on αCTT accessibility in cells (*Figure 5A–D*) suggests that the in vitro assays are missing key factors that regulate αCTT accessibility. We hypothesized that MAPs that recognize the nucleotide and/or conformational state of the microtubule lattice are key to regulating αCTT

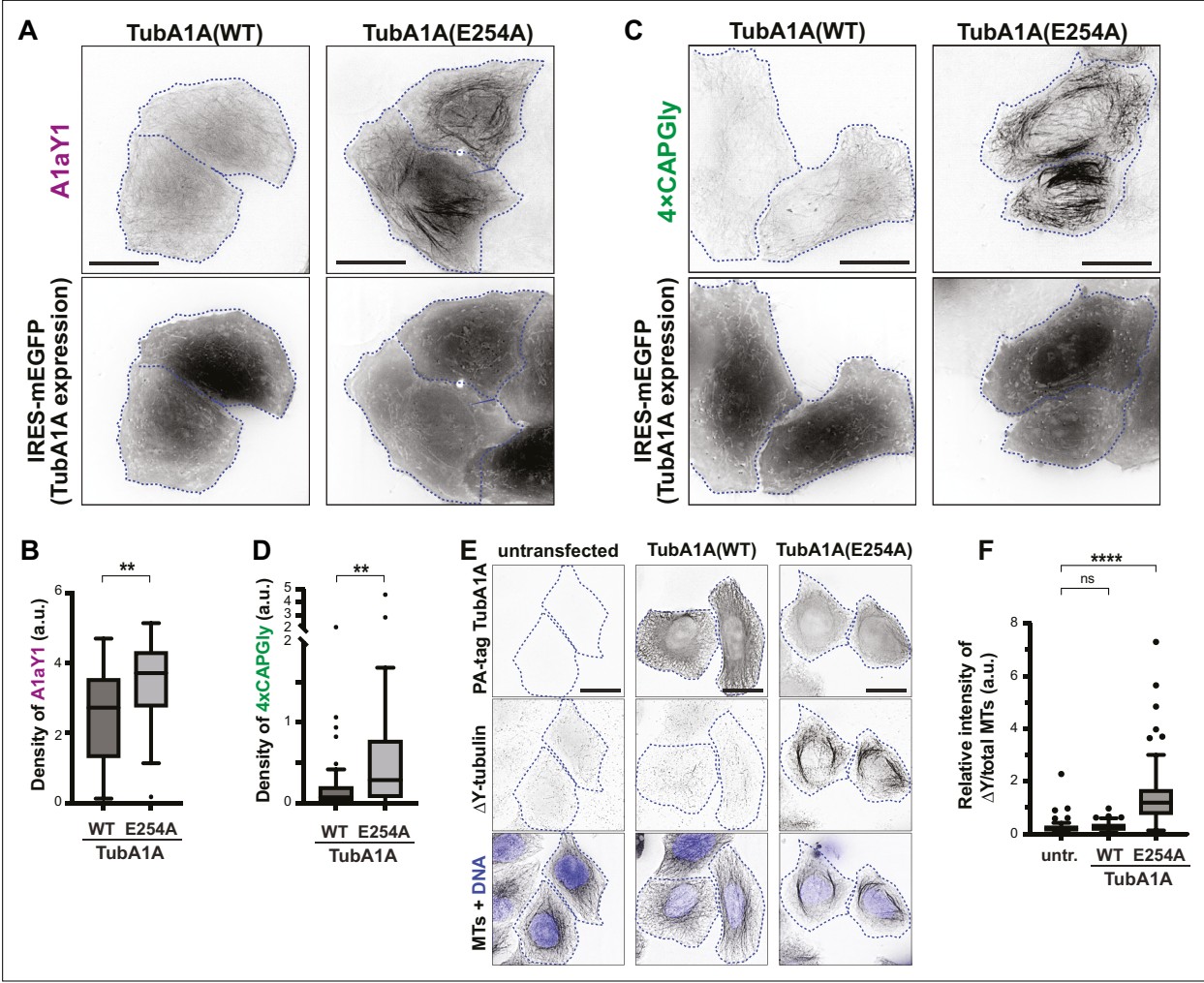

**Figure 5.** GTP-like tubulin state increases Y-αCTT accessibility and detyrosination. (**A–D**) Live-cell imaging of Y-αCTT probes. (**A, C**) Representative images of (**A**) sTagRFP-A1aY1 or (**C**) 4xCAPGly-mSc3 HeLa stable cell lines transiently expressing PA-tagged WT or E254A α-tubulin with an IRES-driven mEGFP protein as a reporter of transfected cells. Cell boundaries are indicated by blue dotted lines. Scale bars: 20 μm. (**B, D**) Quantification of (**B**) A1aY1 or (**D**) 4xCAPGly probe binding to microtubules. The density was measured as the ratio of the skeletonized probe-decorated microtubule length to the total cell area. Data from three independent experiments are presented as Tukey box plots. The box encompasses the 25th to 75th percentiles, with a line at the median. Whiskers show the last data point within 1.5 times the interquartile range. Outliers are plotted as individual points. **: $p < 0.01$ (Mann-Whitney U test). Number of cells analyzed (n) in (**B**): WT = 33, E254A=29 and in (**D**): WT = 42, E254A=50. (**E,F**) Detyrosinated microtubules. (**E**) Representative images of HeLa cells transiently expressing PA-tagged WT or E254A α-tubulin (TubA1A) and then fixed and stained with antibodies against the PA tag, detyrosinated microtubules (ΔY-tubulin), and total microtubules (MTs). Images are shown in inverted grayscale. The nuclei are represented by blue pseudocolor in the bottom panels. Blue dotted lines: boundaries of cells expressing α-tubulin. Scale bars: 20 μm. (**F**) Quantification of the intensity of detyrosination in cells expressing PA-tagged WT or E254A α-tubulin. The fluorescence intensity of detyrosination was measured on a per-cell basis and normalized against the total microtubule intensity. Data from three independent experiments are presented as Tukey box plots. ****: $p < 0.0001$; ns: not significant (Kruskal-Wallis test followed by post-hoc Dunn's multiple pairwise comparisons with the untransfected sample (untr.) as the control). Number of cells analyzed (n): untransfected = 105, WT = 84, E254A=94.

The online version of this article includes the following source data and figure supplement(s) for figure 5:

**Source data 1.** Excel file with fluorescence intensity measurements.

**Figure supplement 1.** Controls for nucleotide-mediated lattice expansion.

**Figure supplement 1—source data 1.** PDF file containing original gels for panels A and B indicating the relevant bands.

**Figure supplement 1—source data 2.** TIFF files of original western blots.

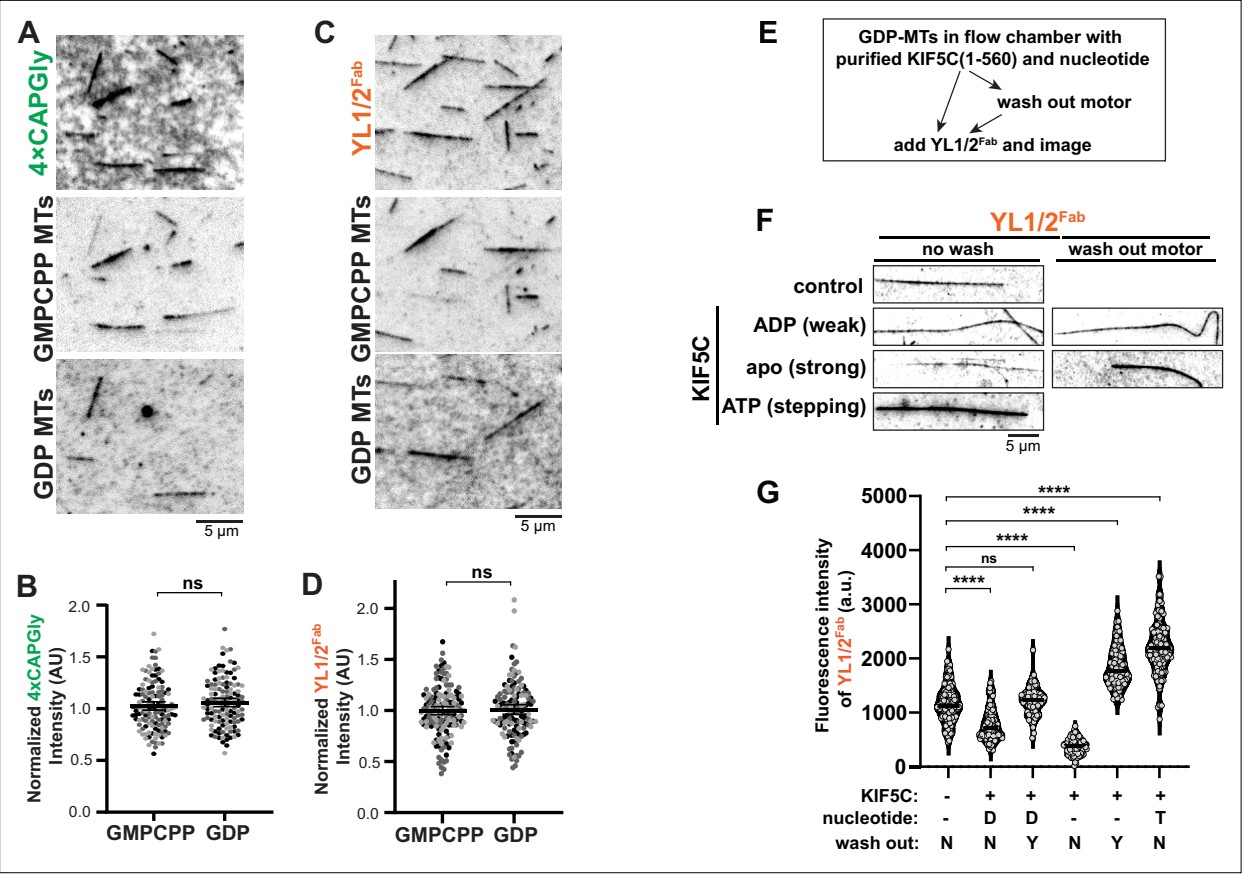

**Figure 6.** MAPs but not nucleotide state promote Y-αCTT exposure in vitro. (**A–D**) Nucleotide state does not determine probe binding to microtubules polymerized in vitro. (**A,C**) Representative images of (**A**) 4xCAPGly-mEGFP or (**C**) YL1/2Fab-GFP probe binding to a mixture containing both AlexaFluor-568 labeled GMPCPP-stabilized microtubules and AlexaFluor-647 labeled GDP microtubules. Scale bars: 5 µm. (**B,D**) Quantification of the fluorescence intensity of (**B**) 4xCAPGly-mEGFP or (**D**) YL1/2Fab-EGFP probe binding per length of microtubule. Data are presented as scatter plots with data from three independent experiments in different shades of gray. ns: not significant (two-tailed, Student's t test). (**E–G**) Stepping KIF5C can increase YL1/2Fab probe binding. (**E**) Flowchart of the in vitro reconstitution assay examining the effect of KIF5C(1-560) stepping on YL1/2Fab binding. (**F**) Representative images of YL1/2Fab-GFP probe binding to GDP-MTs in the absence or presence of KIF5C(1-560) in different nucleotide states. Scale bar: 5 µm. (**G**) Quantification of the mean fluorescence intensity of YL1/2Fab-GFP probe along GDP-MTs under the conditions shown in (**F**). Each spot indicates the probe binding on an individual microtubule. Number of microtubules (n)=74–105 from three independent experiments. ns, not significant, ****p<0.0001 (two-tailed, t-test).

The online version of this article includes the following source data and figure supplement(s) for figure 6:

**Source data 1.** Excel file with fluorescence intensity measurements.

**Figure supplement 1.** Controls for washout of strongly-bound (apo) KIF5C.

---

accessibility. To test this hypothesis in the reconstitution assay, we tested whether the kinesin-1 motor KIF5C could regulate Y-αCTT probe binding. For this, we combined GDP-MTs in a flow chamber with purified KIF5C(1-560) protein and the YL1/2Fab probe protein (*Figure 6E*, left arrow). The ability of KIF5C(1-560) to interact with microtubules and modulate the lattice state can be regulated by nucleotide. In ADP, KIF5C(1-560) binds weakly to microtubules, whereas in the absence of nucleotide (apo state), it is locked in a strong MT-bound state that can induce tubulin expansion. In the presence of ATP, KIF5C(1-560) steps along the lattice and induces transient changes in tubulin conformation (*Peet et al., 2018*; *Shima et al., 2018*).

When KIF5C(1-560) was added in the presence of ADP, the binding of YL1/2Fab to GDP-MTs was reduced as compared to the control condition (no KIF5C) (*Figure 6F and G*). Binding of the YL1/2Fab was even further reduced when KIF5C(1-560) was strongly bound to the GDP-MTs in the apo (no nucleotide) state (*Figure 6F and G*). In contrast, stepping of KIF5C(1-560) in the presence of ATP resulted in a significant increase in YL1/2Fab binding (*Figure 6F and G*).

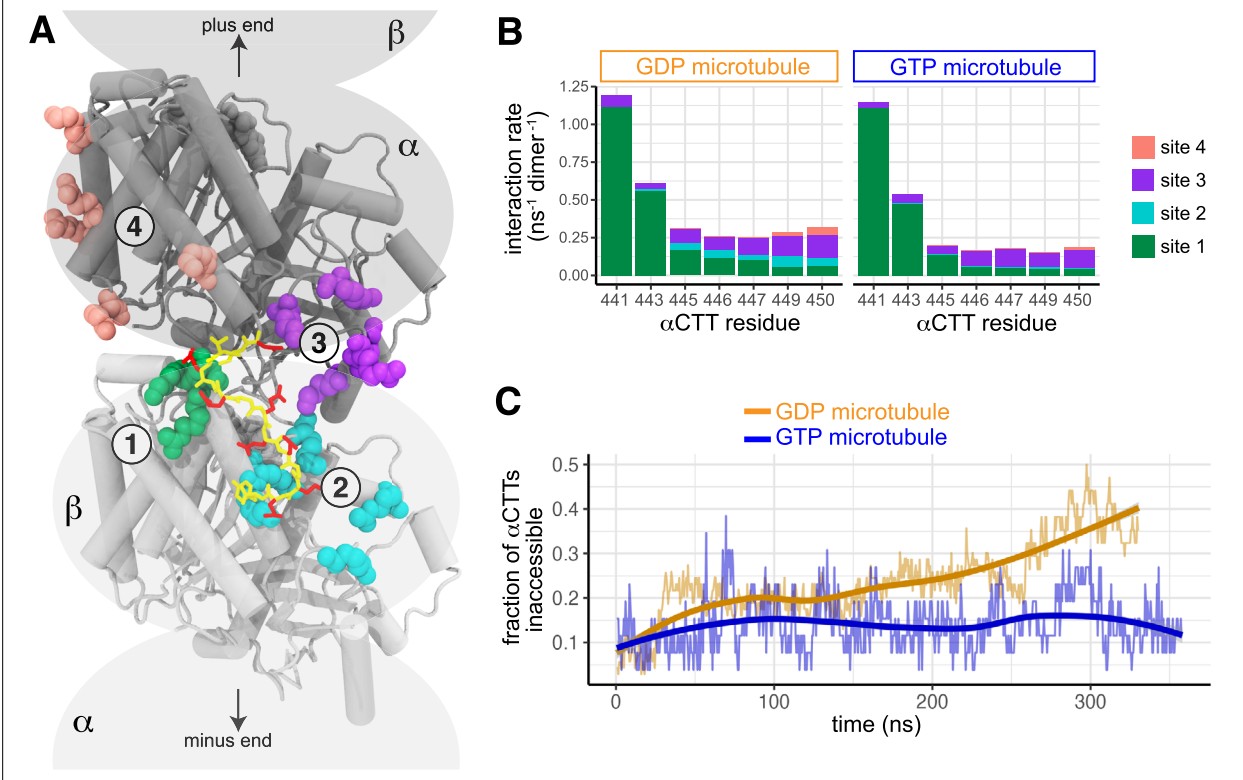

**Figure 7.** The nucleotide state alters Y-αCTT interactions with the microtubule body. (**A**) Representative image from MD simulations identifying four distinct sites where the Y-αCTT interacts with the body of tubulin subunits in the microtubule: sites 1 (green) and 2 (cyan) are on the adjacent β-tubulin along a protofilament (i.e. next tubulin towards the microtubule minus end), whereas sites 3 (purple) and 4 (salmon) are cis-interactions with α-tubulin itself. The tubulin body is shown in cartoon and colored gray. The Y-αCTT is shown in stick and colored yellow with the aspartate and glutamate side chains in red: 438-DSVEGEGEEEGEEY-451. (**B**) Interaction rate of the glutamate residues in the Y-αCTT with the four tubulin body sites for microtubules in the (gold) GDP or (blue) GTP states. (**C**) The fraction of Y-αCTTs that are inaccessible as a function of time for microtubules in the (gold) GDP or (blue) GTP state where inaccessibility is defined as one or more salt bridges formed between glutamates E445-E450 and the interaction sites in the microtubule body.

The online version of this article includes the following figure supplement(s) for figure 7:

**Figure supplement 1.** The Y-αCTT primarily contacts four sites in the tubulin body within a GDP microtubule lattice.

We considered the possibility that Y-αCTT probe binding may be affected by KIF5C binding to the microtubule, for example through steric hindrance. We thus separated KIF5C(1-560) microtubule binding from YL1/2[Fab] microtubule binding using a wash step (*Figure 6E*, right arrows, *Figure 6— figure supplement 1*). After washing out weakly bound KIF5C(1-560) which does not expand the lattice (ADP condition), YL1/2[Fab] binding to GDP-MTs was similar to that of the control (no KIF5C; *Figure 6F and G*). After washing out strongly-bound KIF5C(1-560) (apo state) which does expand the lattice, YL1/2[Fab] binding to GDP-MTs was significantly increased compared to the control (no KIF5C) or weakly-bound KIF5C(1-560) (ADP state; *Figure 6F and G*). These results suggest that strong binding of KIF5C(1-560) to the microtubule lattice, either statically in the apo state or transiently in the ATP state, results in increased Y-αCTT probe binding.

## Molecular dynamics simulations show numerous transient interactions of the αCTT with the GDP microtubule lattice

To probe the accessibility of the αCTT at an atomistic level, we performed molecular dynamics (MD) simulations of microtubules in the GDP state. The simulations identified four distinct sites where the acidic residues of the αCTT form salt bridges with basic residues in the tubulin body (*Figure 7*, *Figure 7—figure supplement 1*). Two of these sites are on the body of the β-tubulin of the adjacent tubulin along the same protofilament (towards the minus end of the microtubule; *Figure 7A*, site 1 (green) and site 2 (cyan)), whereas the other two sites are on the body of the α-tubulin containing

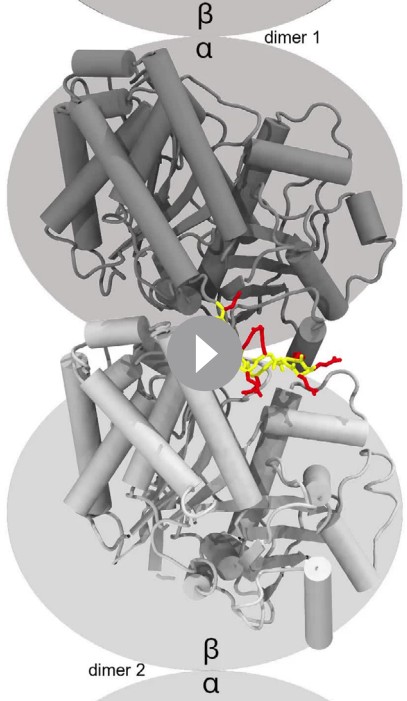

**Video 1.** Movie of Y-αCTT interacting with the tubulin body from molecular dynamics (MD) simulations. Representative movie from MD simulations showing interactions of the Y-αCTT with the tubulin body over 240 ns. The tubulin body is shown in cartoon and colored gray. The αCTT is shown in stick and colored yellow with the glutamate side chains in red. Residues in site 1 (green), site 2 (cyan), and site 3 (magenta) appear as spheres when the αCTT is forming salt bridges with those residues.

https://elifesciences.org/articles/109308/figures#video1

that CTT (*Figure 7A*, site 3 (purple) and site 4 (salmon)). Since there are seven glutamates in the αCTT and five basic residues each in binding sites 2, 3, and 4, there is a very large multiplicity of interacting conformations between the αCTT and the microtubule (*Video 1*).

We next performed MD simulations of microtubules in the GTP state and found that the αCTT formed electrostatic interactions with the same four binding sites in the tubulin body (*Figure 7A*). In both GDP and GTP microtubules, interactions between the αCTT and the tubulin body occurred most frequently between the N-terminal part of the αCTT (E441 and E443) and site 1 of the adjacent β-tubulin (*Figure 7B*), whereas fewer interactions were observed between the more C-terminal residues of the αCTT (residues E445-E450) and the four binding sites (*Figure 7B*). However, the interactions of the more C-terminal part of the αCTT were more susceptible to the nucleotide state of the lattice. In particular, interactions between residues E445-E450 of the αCTT and site 2 of the adjacent β-tubulin were readily observed in the GDP microtubule but nearly abolished in the GTP microtubule (*Figure 7B*). Likewise, interactions between residues E449 and E450 of the αCTT with site 4 of the α-tubulin body were observed in the GDP microtubules but not the GTP microtubules (*Figure 7B*).

We hypothesized that the decreased interactions between the αCTT and the tubulin body in the GTP microtubule would render the αCTT more accessible when tubulin is in the GTP and expanded state. To test this, we calculated how many C-terminal residues of the tail (E445 to Y451) were inaccessible (i.e. bound to one of the tubulin body sites) throughout the course of the simulations. We found that the αCTT residues in the GDP microtubule showed increasing interactions with the tubulin body as a function of time and reached more than 40% inaccessibility after about 300 ns (*Figure 7C*). In contrast, the αCTTs in the GTP microtubule were only about 10% inaccessible throughout the entire simulation time (*Figure 7C*). Taken together, these results support the hypothesis that the αCTT undergoes frequent interactions with the GDP-containing microtubule lattice that render it largely inaccessible to the Y-αCTT probes.

## Discussion
### The αCTT is more accessible along expanded microtubules

In this work, we sought to gain an understanding of the relative accessibility of the αCTT along the microtubule. To this end, we developed three probes to visualize the tyrosinated αCTT: YL1/2[Fab], A1aY1, and 4xCAPGly. We note that our probes are not expected to recognize the two α-tubulin isotypes that do not have a genetically encoded C-terminal tyrosine, TubA4A and TubA8, and play roles in brain development and spermatogenesis, respectively (*Benkirane et al., 2024*; *Diggle et al., 2017*; *Hausrat et al., 2021*). The YL1/2[Fab] and 4xCAPGly proteins can be purified for reconstitution assays, whereas the A1aY1 and 4xCAPGly probes can be expressed in cells, although their optimal use requires live-cell imaging. It is possible that fixation causes the microtubule lattice to expand, but we cannot exclude alternative explanations, for example that fixation disrupts CTT-binding sites on

the tubulin body. We also found that high levels of Y-αCTT probe expression can drive their binding to the microtubule lattice, and it is thus critical to limit examination to cells that express low levels of our probes. This is an important caveat to keep in mind in experiments that utilize transient transfection of these Y-αCTT probes.

We found that the A1aY1 and 4xCAPGly probes showed minimal binding to the microtubule lattice in cells under normal culture conditions. This finding suggests that the αCTT is generally not accessible along the microtubule lattice. Faint microtubule localization is more commonly observed with A1aY1 when compared to 4xCAPGly, which may be due to higher affinity interactions of 4xCAPGly with EB1 and other proteins at the microtubule plus end versus the microtubule lattice. The weak lattice binding of the αCTT probes was surprising as the tubulin CTTs are thought to freely protrude from the microtubule lattice.

Expansion of the microtubule lattice via three orthogonal approaches resulted in an increase in Y-αCTT probe binding: (1) treatment of cells with Taxol (*Figure 2*), (2) expression of MAPs that expand tubulin within the microtubule lattice (expander MAPs; *Figure 3*), and (3) expression of a GTP-locked α-tubulin mutant (E254A, *Figure 5*). These results suggest that accessibility of the αCTT is regulated by the compacted versus expanded state of tubulins within the microtubule lattice. We note that 'expanded' and 'compacted' are used as terms relative to one another rather than defined against a standard microtubule structure (*LaFrance et al., 2022*).

The simplest mechanistic explanation for how tubulin conformational state regulates αCTT accessibility is that the nucleotide state gates αCTT accessibility. However, we find that YL1/2[Fab] and 4xCAPGly probes bind similarly to GMPCPP-MTs vs GDP-MTs polymerized in vitro (*Figure 5*). At first glance, these data conflict with our observation that VASH1/SVBP preferentially detyrosinates microtubules that are in an expanded state (*Yue et al., 2023*). However, it is possible that an expanded lattice promotes activation of VASH1/SVBP, or, alternatively, that engagement of the αCTT with the catalytic site of VASH1/SVBP is optimal on expanded microtubules. Future work is necessary to address these interesting possibilities.

There are several explanations for why αCTT accessibility in reconstitution assays (*Figure 6A–D*) does not correlate with the structural state of the microtubule lattice as observed in cells (*Figure 5*). First, microtubule assembly in vitro may not produce a lattice state resembling that in cells, either due to differences in protofilament number and/or buffer conditions and/or the lack of MAPs during polymerization. Second, the αCTT may switch between accessible and inaccessible states in both GDP- and GTP-microtubules, but the conformation switch rate differs between GDP-microtubules vs GTP-microtubules. In this scenario, Y-αCTT probes would be expected to saturate αCTT-binding sites on both GDP- and GTP-microtubules in vitro. Consistent with this possibility, our MD simulations show that the αCTT undergoes transient interactions with multiple sites along the microtubule lattice. Third, MAPs that bind to or induce an expanded lattice may promote Y-αCTT accessibility. Indeed, we demonstrate that the addition of expander MAPs can increase Y-αCTT accessibility both in cells (*Figure 3*) and for microtubules polymerized under in vitro conditions (*Figure 6E–G*).

## The αCTT is sequestered through interactions with the tubulin body

We propose that the αCTT is sequestered along the microtubule through interactions with the tubulin body in a manner that precludes its recognition by the Y-αCTT probes. Indeed, our MD simulations show that the αCTT undergoes frequent interactions with the bodies of α-tubulin (cis interactions) and β-tubulin (trans interactions). These findings are consistent with an NMR study showing that the αCTT can interact with the body of the α,β-tubulin dimer (*Wall et al., 2016*) and with recent molecular modeling studies suggesting that the CTTs form frequent contacts with themselves and the microtubule body (*Freedman et al., 2011*; *Laurin et al., 2017*; *Luchko et al., 2008*). Furthermore, our data are in agreement with recent modeling work suggesting that CTT-lattice interactions are primarily observed for the αCTT interacting with the β-tubulin of the neighboring dimer towards the minus direction along a single protofilament (*Sahoo and Hanson, 2025*). Our work extends these findings by identifying four primary sites of αCTT-tubulin interactions.

Site 1 involves interactions of the N-terminal glutamate residues of the αCTT (E441, E443) with adjacent β-tubulin basic residues (R390, R391, K392) and our simulations indicate that these interactions are similar between GDP and GTP microtubules. Interestingly, this contact site has been previously observed by cryo-electron microscopy. Specifically, Bodey et al. determined a 9 Å structure of

Taxol-stabilized bovine brain microtubules decorated with the kinesin-5 motor domain and observed density corresponding to the αCTT interacting with β-tubulin of the neighboring tubulin along the longitudinal axis (**Bodey et al., 2009**). More recently, Zehr and Roll-Mecak determined a 2.9 Å structure of GMPCPP-stabilized microtubules polymerized from α1B/βI + α1/βIVb tubulins and observed contacts between V437, S439, and E441 of α-tubulin and residues R390 and R391 of the longitudinally adjacent β-tubulin (**Zehr and Roll-Mecak, 2023**).

In contrast, the interactions at sites 2, 3, and 4 have not been observed in previous modeling or structural studies. These interactions involve salt bridges between acidic residues at the C-terminal end of the αCTT (E445-E450) and clusters of basic residues on the adjacent β-tubulin (site 2: K174, R380, R213, K379, and R306) or on the α-tubulin itself (site 3: K311, R308, K338, R339 and site 4: K112, R123, R156, K163, K430). We note that these interactions are energetically favorable since the salt bridges are a strong enthalpic interaction and the multitude of conformations reduces the entropic penalty. The multiplicity of interactions between the αCTT and the microtubule lattice explains why such interactions are not able to be resolved in structural studies. Moreover, the transient nature of these interactions explains why the Y-αCTT probes show no difference in steady-state binding between reconstituted GMPCPP and GDP microtubules. Importantly, our simulations indicate that the trans interactions with site 2 and the cis interactions with site 4 are reduced in the GTP state, consistent with the αCTT being more accessible along a GTP expanded microtubule lattice.

## Crosstalk between CTTs, MAPs, and PTMs

Overall, our data support a model in which the αCTT is generally not accessible along the GDP-microtubule but can be locally 'exposed' by MAPs. We speculate that, in cells, the microtubule lattice is decorated by MAPs that preferentially recognize the compacted GDP-lattice state and that these compactor MAPs in turn mask the αCTT and/or stabilize αCTT interactions with the microtubule lattice. Lattice expansion, for example through Taxol treatment or overexpression of expander MAPs, would evict compactor MAPs and result in exposure of the αCTTs. Identification of compactor MAPs that help to conceal the αCTT is a major area for future work and represents a significant challenge because our current knowledge of MAPs that bind MTs in a conformation-sensitive manner is limited. Moreover, compactor MAPs may function synergistically or in concert, complicating the identification of MAPs that regulate αCTT accessibility.

An important implication of our work is that the αCTT is not freely available to undergo posttranslational modification. Indeed, we find a strong correlation between the perturbations we used to expand the microtubule lattice and the presence of ΔY-MTs. Taxol treatment expands the microtubule lattice and promotes detyrosination (**Yue et al., 2023**) as does expression of expander MAPs but not compactor MAPs (**Figure 3**) or expression of a GTP-locked tubulin (**Figure 4**). Our findings may also apply to other PTMs of the αCTT, that is polyglycylation and polyglutamylation. This possibility is supported by observations showing that polyglutamylation and detyrosination often co-localize in cells (**Ebberink et al., 2023**), although further pursuit of this possibility is complicated by the fact that the CTTs of both α- and β-tubulin can be modified with glutamylation and we generally lack tools to distinguish between α-tubulin and β-tubulin glutamylation. In addition, it is not known if the accessibility of the βCTT is regulated in a manner similar to the αCTT. Recent findings showing that MATCAP2/TMCP2 trims the CTT of β1-tubulin to generate △3-β1 and that this modified β1-tubulin localizes to specific subcellular structures strongly suggest that post-translational modification of the βCTT is also regulated (**Nicot et al., 2023**). Our work provides a conceptual roadmap to explore this interesting possibility.

A second important implication of our work is how the αCTT can spatially and temporally influence the binding of MAPs to the microtubule. For example, the molecular motors dynein and kinesin-3 KIF13B contain CAP-Gly domains and their initial microtubule interactions (on-rates) are thus facilitated by a tyrosinated αCTT (**Fan and McKenney, 2023**; **McKenney et al., 2016**; **Niederstrasser et al., 2002**). We speculate that these motor proteins preferentially target microtubule segments where expander MAPs have exposed the αCTT. In addition, kinesin-1 expands the microtubule lattice (**Peet et al., 2018**; **Shima et al., 2018**), and as MAP7 is now also thought to expand the lattice (**Figures 3 and 4**; **Shen and Ori-McKenney, 2024**), we speculate that the ability of MAP7 to facilitate the loading of kinesin-1 onto microtubules may involve lattice expansion in addition to protein-protein interactions between MAP7 and kinesin-1 (**Chaudhary et al., 2019**; **Ferro et al., 2022**; **Hooikaas**

*et al., 2019*; *Métivier et al., 2019*; *Monroy et al., 2020*; *Pan et al., 2019*; *Tymanskyj et al., 2018*). Further work will address these and other outstanding questions about the interplay between microtubule lattice state, MAPs, and molecular motors.

# Materials and methods

## Key resources table

| Reagent type (species) or resource | Designation | Source or reference | Identifiers | Additional information |
|---|---|---|---|---|
| Antibody | Anti-β-tubulin, mouse monoclonal, clone E7 | DSHB | Cat# E7; RRID:AB_528499 | 1:1000 IF |
| Antibody | Anti-PA tag, rat monoclonal, clone NZ-1 | FUJIFILM Wako Pure Chemicals | Cat# 016–25861 | 1:1000 WB, 1:500 IF |
| Antibody | Anti-detyrosinated α-tubulin, rabbit monoclonal, clone RM444 | RevMAb Biosciences | Cat# 31-1335-00 | 0.1 µg/ml WB, 0.5 µg/ml IF |
| Antibody | Anti-tyrosinated α-tubulin, rat monoclonal, clone YL1/2 | Bio-Rad | Cat# MCA77G | 1:1000 WB |
| Antibody | rMAb-YL1/2-EGFP recombinant protein | This study | | 1:1000 WB; Verhey lab |
| Antibody | Anti-α-tubulin, mouse monoclonal, clone DM1α | MilliporeSigma | Cat# 05–829 | 1:3000 WB |
| Antibody | Anti-GFP, chicken polyclonal | Aves labs | Cat# GFP-1010 | 1:1000 IF |
| Antibody | Anti-GST, mouse monoclonal | Nacalai USA | Cat# 04435–26 | 1:1000 WB |
| Antibody | Anti-GAPDH, mouse monoclonal, clone G-9 | Santa Cruz | Cat# sc-365062 | 1:2000 WB |
| Antibody | Anti-TagRFP, mouse monoclonal, clone 6A11f | Kerafast | Cat# EFH005 | 1:1000 WB |
| Antibody | Anti-RFP (mScarlet), rabbit polyclonal | Rockland Immunochemicals | Cat# 600-401-379 | 1:1000 WB |
| Antibody | Anti-mouse IgG Alexa Fluor 680 AffiniPure, donkey polyclonal | Jackson ImmunoResearch | Cat# 715-625-150 | 1:500 IF |
| Antibody | Anti-rabbit IgG Alexa Fluor 594, goat polyclonal | Thermo Fisher | Cat# A-11012 | 1:1000 IF |
| Antibody | Anti-rat IgG Alexa Fluor 488 | Thermo Fisher | Cat# A-11006 | 1:1000 IF |
| Antibody | Anti-chicken IgY Alexa Fluor 488 | Thermo Fisher | Cat# A-11039 | 1:1000 IF |
| Antibody | Anti-α-tubulin Alexa Fluor 647, mouse monoclonal, clone DM1α | Millipore Sigma | Cat# 05–829-AF647 | 1:500 IF |
| Antibody | Anti-rat IgG Alexa Fluor 680, goat | Thermo Fisher | Cat# A-21096 | 1:5,000 WB |
| Antibody | Anti-mouse IgG Alexa Fluor 700, goat | Thermo Fisher | Cat# A-21036 | 1:5,000 WB |
| Antibody | Anti-mouse IgG DyLight 800, goat | Thermo Fisher | Cat# SA5-10176 | 1:10,000 WB |
| Antibody | Anti-rat IgG DyLight 800, donkey | Thermo Fisher | Cat# SA5-10032 | 1:5,000 WB |
| Antibody | Anti-rabbit IgG IRDye 800CW, goat | LI-COR | Cat# 926–32211 | 1:10,000 WB |
| Peptide, recombinant protein | Biotinylated porcine brain tubulin | Cytoskeleton | Cat# T333P | |
| Peptide, recombinant protein | HiLyte647 porcine brain tubulin | Cytoskeleton | Cat# TL670M | |
| Peptide, recombinant protein | HiLyte488 porcine brain tubulin | Cytoskeleton | Cat# TL488M | |
| Peptide, recombinant protein | Bovine brain tubulin | Ohi lab | | |
| Peptide, recombinant protein | HeLa S3 tubulin | Ohi and Verhey labs (*Thomas et al., 2025*) | | |
| Peptide, recombinant protein | Alexa Fluor 568 HeLa S3 tubulin | Ohi and Verhey labs (*Thomas et al., 2025*) | | |
| Peptide, recombinant protein | Alexa Fluor 647 HeLa S3 tubulin | Ohi and Verhey labs (*Thomas et al., 2025*) | | |
| Chemical compound, drug | Taxol | Cytoskeleton | Cat# TXD01 | |
| Chemical compound, drug | puromycin | MilliporeSigma | Cat# P8833 | |
| Chemical compound, drug | doxycycline | Thermo Fisher | Cat# BP26531 | |
| Chemical compound, drug | Lipofectamine 2000 | Thermo Fisher | Cat# 11668019 | |
| Chemical compound, drug | Opti-MEM | Thermo Fisher | Cat# 31985070 | |
| Chemical compound, drug | Bovine serum albumin | MilliporeSigma | Cat# A9647 | |

*Continued on next page*

*Continued*

| Reagent type (species) or resource | Designation | Source or reference | Identifiers | Additional information |
|---|---|---|---|---|
| Chemical compound, drug | Casein | MilliporeSigma | Cat# C8654 | |
| Chemical compound, drug | Glycerol | Thermo Fisher | Cat# BP229-4 | |
| Chemical compound, | Glucose | MilliporeSigma | Cat# G7528 | |
| Chemical compound, | Glucose oxidase | MilliporeSigma | Cat# G7141-10KU | |
| Chemical compound, | Catalase | MilliporeSigma | Cat# C3515 | |
| Chemical compound, | GTP | MilliporeSigma | Cat# G8877 | |
| Chemical compound, | GMPCPP | Jena Bioscience | Cat# NU405S | |
| Chemical compound, | ADP | MilliporeSigma | Cat# A2754 | |
| Chemical compound, | Hexokinase | MilliporeSigma | Cat# H5000 | |
| Chemical compound, | Apyrase | MilliporeSigma | Cat# A6535 | |
| Chemical compound, | ATP | MilliporeSigma | Cat# A7699 | |
| Chemical compound, | Janelia Fluor X 554 (JFX554) Halo ligand | Janelia Farms | Cat# JFX554 | |
| Chemical compound, | BSA-biotin | MilliporeSigma | Cat# A8549 | |
| Chemical compound, | NeutrAvidin | Thermo Fisher | Cat# 31000 | |
| Chemical compound, | Fish skin gelatin (FSG) | MilliporeSigma | Cat# G7765 | |
| Chemical compound, | Lysozyme | MilliporeSigma | Cat# L6876 | |
| Chemical compound, | SIGMAFAST protease inhibitor cocktail | MilliporeSigma | Cat# S8830 | |
| Chemical compound, | Benzonase nuclease | MilliporeSigma | Cat# E1014 | |
| Chemical compound, | Biotin | MilliporeSigma | Cat# B4501 | |
| Strain, strain background (*Escherichia coli*) | DH5α | Thermo Fisher | Cat# 18258–012 | |
| Strain, strain background (*Escherichia coli*) | Rosetta2(DE3)pLysS | Novagen | Cat# 71403–3 | |
| Strain, strain background (*Escherichia coli*) | BL21-CodonPlus-RILC | Aligent Technologies | Cat# 230245 | |
| Strain, strain background (*Escherichia coli*) | DH10Bac | Thermo Fisher | Cat# 10361012 | |
| Cell line (*Ceropithecus aethiops*) | COS-7 cells, male kidney fibroblast | ATCC | RRID:CVCL_0224 | |
| Other | Dulbecco's modified Eagle medium (DMEM) | Gibco, Thermo Fisher | Cat# 11960044 | |
| Other | Fetal Clone III | HyClone | Cat# SH3010903 | |
| Other | GlutaMAX (L-alanyl-L-glutamine dipeptide in 0.85% NaCl) | Gibco, Thermo Fisher | Cat# 35050061 | |
| Cell line (*Homo sapiens*) | HeLa Kyoto cells, female | Shuh Narumiya | RRID:CVCL_1922 | |
| Cell line (*Homo sapiens*) | Knock-in HeLa Kyoto cell lines expressing 4xCAPGly-mEGFP, 4xCAPGly-mSc3 or sTagRFP-A1aY1 | This study | | Ohi lab |
| Other | Dulbecco's modified Eagle medium (DMEM) | Gibco, Thermo Fisher | Cat# 11965118 | |
| Other | Fetal Bovine Serum (FBS) | Cytiva | Cat# SH3007103T | |
| Chemical compound, drug | Penicillin-Streptomycin | Gibco, Thermo Fisher | Cat# 15140122 | |
| Cell line (*Homo sapiens*) | HeLa S3 cells, female | ATCC (CCL-2.2) | RRID:CVCL_0058 | |
| Other | Dulbecco's modified Eagle medium (DMEM) | Gibco, Thermo Fisher | Cat# 11965092 | |
| Other | Leibovitz's L-15 Medium | Gibco, Thermo Fisher | Cat# 21083027 | |
| Cell line (*Spodoptera frugiperda*) | Sf9 cells | Thermo Fisher | Cat# 11496015, RRID:CVCL_JX36 | |
| Other | sf900 II SFM medium | Thermo Fisher | Cat# 10902088 | |

*Continued on next page*

*Continued*

| Reagent type (species) or resource | Designation | Source or reference | Identifiers | Additional information |
|---|---|---|---|---|
| Chemical compound, drug | Antibiotic antimycotic | Gibco, Thermo Fisher | Cat# 15240062 | |
| Chemical compound, drug | Cellfectin II | Thermo Fisher | Cat# 10362100 | |
| Other | Grace's insect cell culture medium | Gibco, Thermo Fisher | Cat# 11595–030 | |
| Other | Poly-D-lysine coated glass-bottom dishes | MatTek | Cat# P35GC-1.5–14 C | |
| Other | Prolong Gold | Thermo Fisher | Cat# P36930 | |
| Other | Prolong Diamond | Thermo Fisher | Cat# P36970 | |
| Other | HiLoad 16/600 Superdex 200 prep grade column | Cytiva | Cat# 28989335 | |
| Other | Strep-Tactin XT 4Flow resin | Iba Life Sciences | Cat# 2-5010-002 | |
| Commercial assay, kit | Fluorescent Protein Labeling Kit | Thermo Fisher | Cat# A10235 | |
| Commercial assay, kit | Fab Preparation Kit | Thermo Fisher | Cat# 44985 | |
| Commercial assay, kit | HiPure Plasmid DNA miniprep kit | Thermo Fisher | Cat# K20003 | |
| Recombinant DNA reagent | pN1-4xCAPGly-mEGFP | This study | Lab plasmid | Ohi and Verhey labs |
| Recombinant DNA reagent | pN1-4xCAPGly-mSc3 | This study | Lab plasmid | Ohi and Verhey labs |
| Recombinant DNA reagent | pEM791-4xCAPGly-mEGFP | This study | Lab plasmid | Ohi and Verhey labs |
| Recombinant DNA reagent | pEM791-4xCAPGly-mS3 | This study | Lab plasmid | Ohi and Verhey labs |
| Recombinant DNA reagent | pET15b-6×His-4xCAPGly-mEGFP | This study | Lab plasmid | Ohi and Verhey labs |
| Recombinant DNA reagent | pC1-superTagRFP-A1aY1 | This study | Lab plasmid | Ohi and Verhey labs |
| Recombinant DNA reagent | pEM791-superTagRFP-A1aY1 | This study | Lab plasmid | Ohi and Verhey labs |
| Recombinant DNA reagent | pET15b-6xHis-A1aY1-superTagRFP | This study | Lab plasmid | Ohi and Verhey labs |
| Recombinant DNA reagent | pCAGGS-internal PA-TubA1A-IRES-EGFP | This study | Lab plasmid | Ohi and Verhey labs |
| Recombinant DNA reagent | pCAGGS-internal PA-TubA1A(E254A)-IRES-EGFP | This study | Lab plasmid | Ohi and Verhey labs |
| Recombinant DNA reagent | pCAGGS-internal PA-TubA1A | This study | Lab plasmid | Ohi and Verhey labs |
| Recombinant DNA reagent | pCAGGS-internal PA-TubA1A(E254A) | This study | Lab plasmid | Ohi and Verhey labs |
| Recombinant DNA reagent | pCAGGS-mEGFP-TubA WT | This study | Lab plasmid | Ohi and Verhey labs |
| Recombinant DNA reagent | pCAGGS-mEGFP-TubA WT | This study | Lab plasmid | Ohi and Verhey labs |
| Recombinant DNA reagent | pPA-MAP2-mEGFP | This study | Lab plasmid | Ohi and Verhey labs |
| Recombinant DNA reagent | pPA-MAP7-mEGFP | This study | Lab plasmid | Ohi and Verhey labs |
| Recombinant DNA reagent | pN1-KIFC(1–560,G235A)-mEGFP | This study | Lab plasmid | Ohi and Verhey labs |
| Recombinant DNA reagent | pmEGFP-Tau | This study | Lab plasmid | Ohi and Verhey labs |
| Recombinant DNA reagent | pmEGFP-CAMSAP2 | This study | Lab plasmid | Ohi and Verhey labs |
| Recombinant DNA reagent | pmEGFP-CAMSAP3 | This study | Lab plasmid | Ohi and Verhey labs |
| Recombinant DNA reagent | pCAGGS-mEGFP-Tau | This study | Lab plasmid | Ohi and Verhey labs |
| Recombinant DNA reagent | pCAGGS-MAP2-mEGFP | This study | Lab plasmid | Ohi and Verhey labs |
| Recombinant DNA reagent | pCAGGS- mEGFP-CAMSAP2 | This study | Lab plasmid | Ohi and Verhey labs |
| Recombinant DNA reagent | pCAGGS- mEGFP-CAMSAP3 | This study | Lab plasmid | Ohi and Verhey labs |
| Recombinant DNA reagent | pFastBac1- KIF5C(1-560)-Halo-2xstrepII | This study | Lab plasmid | Ohi and Verhey labs |
| Software, algorithm | Fiji/ImageJ | *Schindelin et al., 2012* | https://fiji.sc/ | |
| Software, algorithm | Fiji Lpx_bilevelThin plugin | *Higaki et al., 2010* | | |
| Software, algorithm | Fiji Ridge Detection plugin | *Wagner and Hiner, 2017* | | |
| Software, algorithm | Fiji EzColocalization plugin | *Stauffer et al., 2018* | | |
| Software, algorithm | AIVIA | DRVision | | |

*Continued on next page*

*Continued*

| Reagent type (species) or resource | Designation | Source or reference | Identifiers | Additional information |
|---|---|---|---|---|
| Software, algorithm | R version 4.3.1 | R Core Team | https://www.R-project.org/ | |
| Software, algorithm | Bio3D | *Grant et al., 2006* | | |
| Software, algorithm | Prism version 10.4.1 | GraphPad Software | https://www.graphpad.com | |
| Software, algorithm | Adobe Illustrator version 29.0 | Adobe | | |

## Y-αCTT probe plasmids

The 4xCAPGly probe consists of rat CLIP1 (CLIP-170) aa 3–484 and is based on a fragment of CLIP-170 previously called H2 (*Arnal et al., 2004*; *Bieling et al., 2008*; *Scheel et al., 1999*). To generate a plasmid for transient mammalian expression of a C-terminally tagged 4xCAPGly probe, the DNA coding for amino acids 3–484 of rat CLIP1 (UniProt Q9JK25) was prepared by PCR amplification using the primers CAPGLYfor1 and CAPGLYrev1 (*Supplementary file 1*). The PCR fragment was joined with BamHI-linearized pmEGFP-N1 or pmScarlet3-N1 (mSc3, *Gadella et al., 2023*) vectors via Gibson Assembly (NEB) to create pN1-4xCAPGly-mEGFP and pN1-4xCAPGly-mSc3 plasmids.

To generate a stable knock-in HeLa-Kyoto cell line expressing 4xCAPGly, the DNA for 4xCAPGly-mEGFP and 4xCAPGly-mS3 was amplified by PCR using primers CAPGLY_for2 and CAPGLY_revGFP or CAPGLY_revSc3 (*Supplementary file 1*). The PCR fragments were inserted via Gibson Assembly (NEB) into a pEM791 vector that was digested with AgeI and BsrGI. The pEM791-4xCAPGly-mEGFP and pEM791-4xCAPGly-mS3 plasmids were used to establish knock-in HeLa cell lines via recombination-mediated cassette exchange (*Khandelia et al., 2011*) in order to express 4xCAPGly-mEGFP or 4xCAPGly-mS3 proteins in a doxycycline-inducible manner.

To generate a plasmid for bacterial expression of a 6×His-tagged 4xCAPGly-mEGFP probe, the DNA coding for 4×CAPGly-mEGFP was PCR amplified from the pmEGFP-N1-4xCAPGly plasmid using primers CLIP_for3 and CLIP_rev3 (*Supplementary file 1*). The PCR fragment was joined with a BamHI-linearized pET15b vector via Gibson Assembly (NEB).

To generate a plasmid for transient mammalian expression of the superTagRFP-tagged A1aY1 probe, a 232 bp fragment of A1aY1 lacking the start codon was synthesized (IDT) and PCR amplified using the primers A1aY1_for1 and A1aY1_rev1 (*Supplementary file 1*). The PCR fragment was inserted into an EcoRI-linearized psuperTagRFP-C1, in which mEGFP of pmEGFP-C1 vector had been replaced with superTagRFP (sTagRFP, *Mo et al., 2020*) using Gibson Assembly, resulting in the plasmid pC1-superTagRFP-A1aY1.

To generate a stable knock-in HeLa-Kyoto cell line expressing superTagRFP-A1aY1, the coding sequence was amplified from pC1-superTagRFP-A1aY1 using the primers A1aY1_for2 and A1aY1_rev2 (*Supplementary file 1*) and inserted into a pEM791 vector that was linearized by digestion with AgeI and BsrGI. The resulting pEM791-superTagRFP-A1aY1 plasmid was used to establish a knock-in HeLa cell line via recombination-mediated cassette exchange (*Khandelia et al., 2011*) in order to express superTagRFP-A1aY1 protein in a doxycycline-inducible manner.

To generate a plasmid for bacterial expression of the A1A1 probe, we placed the fluorophore at the C-terminus of A1aY1 based on a previously published study (*Kesarwani et al., 2020*). Fragments of superTagRFP and A1aY1 were separately PCR-amplified with primers A1aY1_for3 and A1aY1_rev3, and sTagRFP_for1 and sTagRFP_rev1 (*Supplementary file 1*), respectively. These two fragments were inserted then into BamHI-linearized pET15b vector via Gibson Assembly, which yielded the construct pET15b-6xHis-A1aY1-superTagRFP. This construct was used to produce the recombinant fusion protein 6xHis-A1aY1-sTagRFP.

## Tubulin plasmids

The plasmid for expressing PA-tagged human TubA1A WT and E254A constructs was generated by replacing the 6xHis tag inserted between Ile42 and Gly43 in the flexible loop of TubA1A in the vectors pJM546 and pJM602 (kindly provided by Jeff Moore, University of Colorado Anschutz Medical Center) with a PA-tag (GVAMPGAEDDVV). The resulting plasmids contain PA-tagged TubA1A followed by an internal ribosome entry site (IRES) driving the expression of an EGFP reporter. The template plasmids were PCR amplified using phosphorylated primers His-PA_for and His-PA_rev (*Supplementary file*

1), and ligated, yielding pCAGGS-internal PA-TubA1A-IRES-EGFP. For immunofluorescence studies, constructs without the IRES-GFP component were used. These versions were generated by amplifying the full vectors excluding the IRES-GFP region using phosphorylated primers IRES-GFP_removal_ for and IRES-mEGFP_removal_rev (*Supplementary file 1*), followed by ligation to re-circularize the vectors.

For GFP-tagged TubA1A expression, untagged TubA1A was amplified using primers mEGFP-TubA1A_for and mEGFP-TubA1A_rev, and inserted into a pCAGGS vector that had been linearized with primers IRES-GFP_removal_for and pCIG2_rev (*Supplementary file 1*). The insertion was performed via Gibson assembly, yielding the pCAGGS-mEGFP-TubA WT construct.

## MAP plasmids

MAP2 and MAP7 were tagged at their N-termini with the PA tag and at their C-termini with mEGFP (PA-MAP2-mEGFP and PA-MAP7-mEGFP). The coding sequence of human MAP2 (UniProt P11137-4) was amplified from cDNA (Horizon Discovery; Clone ID 5223046) using primers MAP2_for and MAP2_rev (*Supplementary file 1*). The coding sequence of mouse MAP7 (UniProt O88735-2) was amplified from pCAGG-MAP7(FL)-mCherry (*Tymanskyj et al., 2018*) using primers MmMAP7_for and MmMAP7_rev for MAP7 (*Tymanskyj et al., 2018*). The PCR products were used as inserts in a Gibson assembly reaction with pPA-mEGFP-N1-EML2-S plasmid (*Hotta et al., 2022*) that was linearized and PCR-amplified with primers PA-mEGFP-N1_for1 and PA-mEGFP-N1_rev1 (*Supplementary file 1*).

For the Kif5C$^{rigor}$ construct, the rat Kif5C(1–560,G235A) sequence (UniProt P56536) was amplified from pN1-KIFC(1–560,G235A) using primers Kif5C_for and Kif5C_rev (*Supplementary file 1*). The resulting PCR product was used as an insert in a Gibson assembly with the pmEGFP-N1 vector linearized and PCR-amplified with primers PA-mEGFP-N1_for2 and PA-mEGFP-N1_rev2.

Tau, CAMSAP2, and CAMSAP3 were tagged at their N-termini with mEGFP. Human tau (UniProt P10636-6) was PCR-amplified from cDNA (Horizon Discovery, Clone ID 40007445) using primers tau_for and tau_rev (*Supplementary file 1*). Human CAMSAP2 (UniProt Q08AD1-1) was PCR-amplified from pEGFP-C1-CAMSAP2 (*Yue et al., 2023*) using primers CAMSAP2_for and CAMSAP2_rev (*Supplementary file 1*). Human CAMSAP3 (UniProt Q9P1Y5-2 with N-terminal 19 amino acid deletion) was PCR-amplified from cDNA (Horizon Discovery, Clone ID, 3868695) using primers CAMSAP3_for and CAMSAP3_rev (*Supplementary file 1*). The PCR products were used as inserts in a Gibson assembly reaction with the pPA-mEGFP-C1 vector (*Hotta et al., 2022*) linearized by EcoRI digestion.

In some cases, transient expression of these MAPs suppressed the expression of Y-αCTT probes in the knock-in HeLa cell lines. To mitigate this, mEGFP-tagged tau, MAP2, CAMSAP2, and CAMSAP3 were further transferred into pCAGGS vectors containing a β-actin promoter. The pCAGGS vector was PCR-amplified with primers pCAGGS_for and pCAGGS_rev (*Supplementary file 1*), and the inserts were added by Gibson assembly. The primers mEGFP_MAP_for and mEGFP_MAP_rev were used to PCR-amplify inserts of mEGFP-tau, mEGFP-CAMSAP2, and mEGFP-CAMSAP3, whereas the primers mEGFP_MAP_for and PA-MAP2-mEGFP_rev were used to PCR-amplify the insert of MAP2-mEGFP. All plasmids were validated by Sanger sequencing.

## Generation of rMAb-YL1/2-EGFP antibody and Fab

We determined the protein sequences of the immunoglobulin G (IgG) heavy and light chains of the rat monoclonal anti-tyrosinated α-tubulin antibody YL1/2 using mass spectrometry (Rapid Novor). As leucine and isoleucine have the same mass, their identities were distinguished by further fragmentation at the Cβ-Cγ bond to release a propyl (leucine) or ethyl (isoleucine) group (W-ion determination, Rapid Novor *Zhokhov et al., 2017*). The Clothia method was used to identify the three complementarity determining regions (CDRs) within the variable regions of each light chain and heavy chain (*Chothia and Lesk, 1987*). CDRs are hypervariable loops that form the antigen binding interface and thus determine antibody specificity and affinity. The surrounding sequences (framework regions) dictate and maintain the folded state of the variable domain such that the CDRs are properly displayed on one side of the folded protein (*Zhu et al., 2025*).

To generate a recombinant monoclonal antibody (rMAb), the coding sequences of the HC and LC were tagged at the N-terminus with a signal sequence (MGWSCIILFLVATATGVHS) for entry into the secretory pathway. The coding sequence of the LC was additionally modified by tagging the C-terminus with a GGGGS linker followed by EGFP. Recombinant YL1/2-EGFP protein was produced

using the CHO-Express system (Genscript). A Fab fragment of the YL1/2-EGFP protein was prepared using the Fab Preparation Kit (Thermo Fisher Scientific; Cat# 44985). Briefly, recombinant YL1/2-EGFP protein was digested with papain for 5 hr at 37 °C followed by negative purification via a Protein A column. Flow-through fractions from the Protein A column were combined and concentrated using an Amicon Ultra-10K ultrafiltration device (Millipore, Cat# UFC801024). The YL1/2$^{Fab}$-EGFP (dissolved in 1×PBS) was aliquoted, flash frozen, and stored at –80 °C until use.

## Protein expression and purification

The 6×His-tagged 4xCAPGly-mEGFP protein was expressed in BL21-CodonPlus-RILC *E. coli* cells. Bacterial cultures were induced with 0.5 mM IPTG at 18°C for 18 hr. Purification was performed based on *Bieling et al., 2008*. Cell pellets were resuspended in lysis buffer [50 mM KPi (pH 7.5), 500 mM NaCl, 1 mM MgCl$_2$, 1 mM BME, 1 mM PMSF, 1.0 mg/ml lysozyme, 1 × SIGMAFAST, and Benzonase nuclease], sonicated, and clarified by centrifugation. The lysate was applied directly to Ni-NTA resin that was pre-equilibrated with lysis buffer. Once the protein was bound, the column was washed with 10 column volumes of low imidazole wash buffer [50 mM KPi (pH 7.5), 500 mM NaCl, 1 mM MgCl$_2$, 8.5 mM Imidazole (pH 7.4), and 1 mM BME] followed by an additional wash with 3 column volumes of high imidazole wash buffer [50 mM KPi (pH 7.5), 500 mM NaCl, 1 mM MgCl$_2$, 125 mM Imidazole (pH 7.4), and 1 mM BME]. Proteins were eluted with elution buffer [50 mM KPi (pH 7.5), 500 mM NaCl, 1 mM MgCl$_2$, 300 mM Imidazole (pH 7.4), and 1 mM BME]. Fractions containing the protein were combined and gel filtered over a HiLoad 16/600 Superdex 200 column that was equilibrated with 50 mM KPi (pH 7.5), 150 mM NaCl, 1 mM MgCl$_2$, and 1 mM BME. Fractions containing the protein were pooled, concentrated with an appropriately sized MWCO centrifugal concentrator, aliquoted, flash frozen, and stored at -80°C until use.

The 6×His-sTagRFP-A1aY1 protein was expressed in Rosetta 2(DE3) pLysS *E. coli* cells. The 6×His-sTagRFP-A1aY1 protein was purified similarly to the 4xCAPGly proteins with the following changes: the cells were lysed in lysis buffer (50 mM KPi [pH 7.5], 500 mM NaCl, 20 mM imidazole, 1 mM PMSF, 1 × SIGMAFAST, and Benzonase). The low imidazole wash step was omitted and the column was washed with Ni-NTA wash buffer (50 mM KPi [pH 7.5], 500 mM NaCl, 20 mM imidazole, 1 mM MgCl2, 1 mM ATP, and SIGMAFAST) and protein was eluted from the column with elution buffer (50 mM KPi [pH 7.5], 500 mM NaCl, 200 mM imidazole). The protein was separated by gel filtration using buffer (10 mM Hepes, 300 mM KCl, 1 mM DTT [pH 7.5]). Protein concentration was determined via Bradford protein assay.

The KIF5C(1-560) protein was expressed in Sf9 cells. The cells were cultured in suspension with serum-free sf900 II SFM medium supplemented with antibiotic antimycotic in flasks at 28°C in a non-CO$_2$ nonhumidified incubator with an orbital shaker platform set at 110 rpm. The KIF5C(1-560) protein consists of rat KIF5C aa 1–160 fused to Halo tag and a dual StrepII tag [KIF5C(1-560)-Halo-2xstreptII]. The construct was subcloned into the pFastBac1 vector via Gibson assembly. Baculovirus was generated according to the Bac-to-Bac system (Thermo Fisher). In brief, plasmids were transformed into DH10Bac *E. coli* to generate recombinant bacmids. Bacmid DNA was isolated with the HiPure Plasmid DNA miniprep kit and confirmed by PCR analysis. Recombinant bacmid DNA was transfected into Sf9 cells using Cellfectin II. After 7 d, the supernatant containing P1 baculovirus was collected and clarified by centrifugation at 3000 rpm for 3 min at 4°C. The baculovirus was amplified by successive infection of Sf9 cells to generate P2 and P3 baculoviruses. Baculovirus-containing supernatants were stored at 4°C in the dark. To purify KIF5C(1-560)-Halo-StrepII protein, Sf9 cells were infected with 3% P3 baculovirus (vol/vol). After 3 days, the cells were harvested by centrifugation for 15 min at 3000 rpm at 4°C. The pellet was washed once with PBS and resuspended in ice-cold lysis buffer (200 mM NaCl, 4 mM MgCl$_2$, 0.5 mM EDTA, 1 mM EGTA, 0.5% igepal, 7% sucrose, and 20 mM imidazole-HCl, pH 7.5) supplemented with 2 mM ATP, 1 mM PMSF, 5 mM DTT, and protease inhibitor cocktail. After 30 min incubation on ice, the lysates were clarified by ultracentrifugation for 20 min at 20,000 rpm in F12−8x50 y rotor (Sorvall 3421), and the supernatants were incubated with strep-Tactin beads for 1 hr at 4°C with rotation. The beads were transferred to a PD-10 column and washed with wash buffer (150 mM KCl, 25 mM imidazole-HCl, pH 7.5, 5 mM MgCl$_2$, 1 mM EDTA, and 1 mM EGTA) supplemented with 1 mM PMSF, 3 mM DTT, 3 mM ATP, and protease inhibitor cocktail. Bound proteins were eluted in 6x0.5 mL fractions with elution buffer (25 mM KCl, 25 mM imidazole-HCl, pH 7.5, 5 mM EGTA, 2 mM MgCl$_2$, 2 mM DTT, 0.1 mM ATP, 1 mM PMSF, protease inhibitor cocktail and

10% glycerol) supplemented with 50 mM biotin. The fractions were separated by SDS-PAGE and fractions containing KIF5C(1-560) were combined and dialyzed in dialysis buffer (25 mM imidazole-HCl, pH 7.5, 25 mM KCl, 5 mM EGTA, 2 mM MgCl$_2$, 2 mM DTT, 0.1 mM ATP and 10% glycerol). After 2 hr, the buffer was changed with fresh dialysis buffer and dialyzed overnight at 4°C to remove biotin from the sample. The protein was concentrated by centrifugation, and aliquots were snap-frozen in liquid nitrogen and stored in –80°C until further use.

## Western blot analysis

The specificity of the Y-αCTT probes for the tyrosinated αCTT was validated utilizing GST-αCTT fusion proteins in which glutathione-S-transferase (GST) was fused to human TubA1A αCTT sequences representing tyrosinated (Y=SVEGEGEEEGEEY), detyrosinated (ΔY=SVEGEGEEEGEE), or ΔC2 (Δ2=SVEGE-GEEEGE) tails. GST-αCTT proteins were expressed and purified from bacteria as previously described (*Hotta et al., 2022*; *Hotta et al., 2023*). GST-αCTT proteins (250 ng) were separated by SDS-PAGE and transferred on to nitrocellulose membranes. The membranes were blocked with 3% skim milk in 1×TBS supplemented with 0.1% tween-20 (TBST blocking buffer) for 1 hr at room temperature followed by incubation with primary and secondary antibodies (Key Resource Table) for 1 hr at room temperature. Alternatively, membranes were incubated with recombinant proteins rMAb-YL1/2-EGFP (2 mg/mL), YL1/2$^{Fab}$-EGFP (4 mg/mL), or 4×CAPGly-mEGFP protein (100 nM) in blocking buffer for 30 min at room temperature.

HeLa cell lysates were prepared by resuspending cell pellets in lysis buffer (6 mM Na$_2$HPO$_4$, 4 mM NaH$_2$PO$_4$, 2 mM EDTA, 150 mM NaCl, 1% NP40 and protease inhibitors) followed by a brief sonication and clarification via centrifugation. Cell lysates (15 µg) were separated by SDS-PAGE and transferred to nitrocellulose membranes. The membranes were blocked with 5% skim milk in PBS supplemented with 0.5% tween-20 (PBST blocking buffer), and primary antibody incubation was carried out at 4 °C overnight. Primary antibodies and dilutions are listed in the Key Resources Table. After incubation with primary antibodies, the membranes were washed three to five times for 5 min each with the TBST blocking buffer or PBST and then incubated with the corresponding secondary antibodies for 1 hr at room temperature. Secondary antibodies and dilutions are listed in the Key Resources Table. Membranes were washed three to five times for 5 min each. Fluorescence signals were detected with the Azure 600 imaging system.

## Mammalian cell maintenance and transfection

COS-7 cells were grown in Dulbecco's modified Eagle medium (DMEM) supplemented with 10% (vol/vol) Fetal Clone III and 2 mM GlutaMAX. HeLa Kyoto cells (female *Homo sapiens* (RRID:CVCL_1922)) were maintained in DMEM containing 10% fetal bovine serum (FBS) and 1% Penicillin-Streptomycin. HeLa Kyoto knock-in cell lines expressing 4xCAPGly-mEGFP, 4xCAPGly-mSc3, and sTagRFP-A1aY1 in a doxycycline-inducible manner were maintained in DMEM containing 10% FBS, 1% Penicillin-Streptomycin, and 1 mg/mL puromycin. The expression of each transgene in each knock-in cell line was induced via the addition of 2 µg/mL doxycycline. HeLa S3 cells were grown in suspension in DMEM supplemented with 10% (vol/vol) Fetal Clone III Serum, 2 mM GlutaMAX, and 1% Penicillin/Streptomycin.

All cell lines were maintained in the presence of 5% CO$_2$ at 37°C. All cell lines were screened and found negative for mycoplasma contamination. HeLa and COS-7 cells were transfected with Lipofectamine 2000 according to the manufacturer's instructions. Briefly, 0.5–1 µg of plasmid DNA was diluted in 250 µL of Opti-MEM. Following a 5 min incubation at room temperature, 250 µL of Opti-MEM containing 5 µL of Lipofectamine 2000 was added to the plasmid DNA and incubated for 20 min at room temperature. The entire reaction mixture was added directly to the dish of cells containing 1 mL of Opti-MEM and incubated for 3.5 hr at 37°C before exchanging the media to DMEM supplemented with 10% FBS and 1% Penicillin-Streptomycin.

## Total internal reflection fluorescence microscopy

All assays used HeLa microtubules. For this, tubulin was purified from HeLa S3 cells using a GST-TOG column and polymerized into microtubules as described (*Thomas et al., 2025*). A flow cell (~10 µL volume) was assembled by attaching a clean #1.5 coverslip (Thermo Fisher Scientific) to a glass slide (Thermo Fisher Scientific) with two strips of double-sided tape. All imaging was performed at room

temperature. In vitro assays were performed on an inverted Nikon Ti-E/B total internal reflection fluorescence microscope with a perfect focus system, a 100×1.49 NA oil immersion TIRF objective, three 20 mW diode lasers (488 nm, 561 nm, and 640 nm), and EMCCD camera (iXon+DU879; Andor). Image acquisition was controlled using Nikon Elements software.

To validate the sensitivity of the Y-αCTT probes to the tyrosinated vs detyrosinated state of α-tubulin, AlexaFluor 488- or 647-labeled microtubules were generated. Taxol-stabilized microtubules were incubated in a flow chamber with cell lysate overexpressing VASH1 and SVBP for 15 min to generate ΔY-microtubules (*Thomas et al., 2025*; *Yue et al., 2023*). The chambers were washed with 0.1 mg/mL casein and 10 μM Taxol in BRB80 to remove VASH1/SVBP enzyme. Then an imaging mixture containing YL1/2[Fab]-EGFP (66 nM) or 4xCAPGly-mEGFP (15 nM) proteins in 10 μM Taxol in BRB80 supplemented with 0.1 mg/mL casein and oxygen scavengers (1 mM DTT, 1 mM $MgCl_2$, 10 mM glucose, 0.2 mg/mL glucose oxidase, and 0.08 mg/mL catalase) was flowed into the chambers and imaged.

To compare Y-αCTT probe binding to GMPCPP vs GDP microtubules, microtubules were polymerized from HeLa S3 tubulin in the presence of GMPCPP or GTP and in the presence of AlexaFluor 568-labeled HeLa tubulin or AlexaFluor 647-labeled HeLa tubulin (*Thomas et al., 2025*). The chambers were prepared by first flowing in a recombinant anti-TubB3 Tuj1 antibody (gift from Jeffrey Moore, University of Colorado Anschutz Medical Center) diluted 1:50 in BRB80 and incubating for 5–10 min. Chambers were washed with 25% glycerol in BRB80, and then a mixture of GMPCPP and GDP microtubules was added and incubated for 5–10 min. The chambers were washed with 25% glycerol/BRB80, and then YL1/2[Fab]-EGFP (final concentration of 66 nM) or 4xCAPGly-mEGFP (final concentration of 15 nM) proteins were added in 25% glycerol/BRB80 supplemented with 0.1 mg/mL casein and oxygen scavengers (1 mM DTT, 1 mM $MgCl_2$, 10 mM glucose, 0.2 mg/mL glucose oxidase, and 0.08 mg/mL catalase).

To examine whether the binding behavior of YL1/2[Fab]-EGFP on glycerol-stabilized GDP-MTs is regulated by KIF5C(1-560)-Halo protein, GDP-MTs were assembled from HeLa S3 tubulin and stabilized in 25% glycerol/BRB80. The microtubules were introduced into a flow cell and incubated for 3 min at room temperature to allow for nonspecific adsorption to the coverslips. After washing with blocking buffer (1 mg/mL casein and 25% glycerol in P12 buffer (12 mM PIPES/KOH pH 6.8, 2 mM $MgCl_2$, 1 mM EGTA)), the flow cell was infused with imaging buffer (P12 buffer supplemented with 25% glycerol, 0.3 mg/mL casein, and oxygen scavenger mix) with YL1/2[Fab]-EGFP (52 nM) under four conditions: (1) no KIF5C, (2) weakly-bound KIF5C (100 nM KIF5C, 2 mM ADP, 2 units/mL hexokinase), (3) strongly-bound KIF5C (100 nM KIF5C, 6 units/mL apyrase) or (4) stepping KIF5C (100 nM KIF5C, 2 mM ATP). After 3 min, the flow cell was then sealed with molten paraffin wax and imaged.

For the washout experiments, the flow cell was infused with imaging buffer containing weakly-bound KIF5C(1-560)-Halo[554] protein (100 nM KIF5C, 2 mM ADP, 2 units/mL hexokinase) or strongly-bound (apo) KIF5C(1-560)-Halo[554] (100 nM KIF5C, 6 units/mL apyrase). After 3 min, KIF5C was removed by washing the flow cell with blocking buffer supplemented with 3 mM ATP and 300 mM KCl (*Figure 6—figure supplement 1*) based on the experiments in *Shima et al., 2018*. The washing buffer was immediately exchanged to imaging buffer supplemented with 52 nM YL1/2[Fab]. The flow cell was then sealed with molten paraffin wax and imaged. Fluorescence intensities along the microtubules were measured using Fiji/ImageJ (width = 3 pixels), and the fluorescence intensity of an adjacent region was subtracted to account for background noise.

## Live-cell imaging of tail and lattice sensors

For live cell imaging, cells were plated onto poly-D-lysine coated glass-bottom dishes 24–48 hr before imaging and transfected with plasmids 18–24 hr prior to imaging. HeLa stable cell lines were induced for 15 hr with 2 μg/mL doxycycline prior to imaging to express superTagRFP-A1aY1 or 4xCAPGly-mEGFP proteins. In preparation for imaging, each dish was washed with 1 mL of warm Leibovitz's L-15 Medium supplemented with 10% FBS and 1% Penicillin-Streptomycin. Cells were then incubated in imaging medium for 10 min prior to imaging.

To assess changes in Y-αCTT probe localization, cells exhibiting similar levels of fluorescence were identified prior to imaging. Single optical sections were imaged. The localization of each Y-αCTT probe was monitored before treatment and upon treatment with 10 μM Taxol or 0.3% DMSO (vehicle control). After 15 min, the same optical sections of each cell were imaged to directly compare the

localization and intensity of each probe along the microtubule lattice in a compacted (before) and expanded (after) state. All cells were imaged and maintained at 37°C using a DeltaVision Elite microscope system equipped with an Olympus Plan Apo N 60×1.42 NA oil immersion objective.

For live imaging of COS-7 cells, cells were plated onto glass-bottom dishes 48 hr prior to imaging. Cells were transfected with 0.5 µg of pEGFP-N1 and 0.2 µg of Y-αCTT probe plasmid 24 hr prior to imaging. Cells were transferred to warm Leibovitz's L-15 medium for imaging. Images were acquired on an inverted epifluorescence microscope (Nikon TE200E) with a 60x, 1.4 NA oil-immersion objective, TokaiHit stage-top incubator (INUG2A-GILCS) set to 37°C, and an Orca Flash4 OLT digital CMOS camera (Hamamatsu).

## Cell fixation and immunofluorescence

COS-7 cells plated onto glass coverslips were rinsed with PBS+ (PBS with 0.9 mM CaCl$_2$, 0.5 mM MgCl$_2$), then fixed and permeabilized simultaneously in pre-chilled methanol (MeOH) for 10 min at −20°C. Cells were then either washed twice with PBS+, rinsed with ddH$_2$O, dabbed on a kimwipe to remove excess H$_2$O, and mounted onto a slide with Prolong Gold or washed twice with PBS+, blocked briefly with 0.2% fish skin gelatin (FSG) in PBS+ and subjected to immunostaining. Note that soluble EGFP used as a control in these experiments is not preserved with MeOH fixation. Mouse anti-β-tubulin #E7 primary antibody was diluted in 0.2% FSG in PBS+ and applied to cells for 1 hr. Cells were washed three times with 0.2% FSG in PBS+ and then subjected to secondary antibody staining for 1 hr with Alexa Fluor 680-conjugated anti-mouse in 0.2% FSG in PBS+. Finally, cells were washed an additional three times with 0.2% FSG in PBS+ and then twice with PBS+. Coverslips were dipped in ddH$_2$O, dabbed on a kimwipe to remove excess water, and mounted with Prolong Gold.

HeLa cells were cultured on coverslips 24 hr prior to transfection. Approximately 20 hr post-transfection, cells were fixed with methanol for 10 min at −20°C. Following fixation, cells were rehydrated in TBS supplemented with 0.1% Triton X-100 (TBS-Triton), and then blocked for 15 min with 2% BSA in TBS-Triton. All primary and secondary antibodies were diluted in 2% BSA in TBS-Triton. Detyrosinated microtubules were stained with a recombinant rabbit monoclonal anti-detyrosinated α-tubulin antibody for 1 hr followed by anti-rabbit Alexa Fluor 594 for 30 min. Overexpressed MAPs were stained with anti-GFP antibody for 1 hr, followed by anti-chicken IgY Alexa Fluor 488 for 30 min. PA-tagged TubA was stained with a rat monoclonal anti-PA-tag antibody for 1 hr and the anti-rat IgG Alexa Fluor 488 for 30 min. Finally, total microtubules were stained with anti-α-tubulin antibody conjugated with Alexa Fluor 647 for 30 min. DNA was counterstained with Hoechst, and cells were mounted with Prolong Diamond. Images were obtained with a DeltaVision microscope equipped with an Olympus Plan Apo N 60x/1.42 oil immersion lens. Images were subsequently deconvolved, and single optical sections are presented.

## Quantification of αCTT probes and detyrosination in live cells

Microtubule-bound A1aY1 or CAPGly sensors in Taxol-treated or TubA-expressing cells were quantified as the total length of microtubule regions labeled by Y-αCTT probe per unit cell area. Except for sTagRFP-A1aY1 in the Taxol treatment (see below), we measured microtubule density using an image processing method developed recently (*Horiuchi et al., 2025*). First, to enhance the visualized microtubule signal, raw microscopic images were subjected to 2D segmentation using a deep learning-based function in the image analysis software AIVIA. This deep learning model, based on RCA-UNet for image transformation, was trained on a dataset consisting of raw images acquired using various microtubule-targeting probes and their corresponding binarized images, in which visualized microtubule regions were manually segmented (*Horiuchi et al., 2025*). The enhanced images were then binarized using Otsu's thresholding in ImageJ. This segmentation process was designed to apply a strict intensity threshold, isolating high-intensity, localized binding sites from the background noise, and thus inherently excluding low-intensity binding. The binary images were subsequently skeletonized using the ImageJ plug-in Lpx_bilevelThin (*Higaki et al., 2010*). Cell regions were manually segmented using the Polygon Selection tool in ImageJ. Finally, microtubule density (referred to as occupancy) was calculated as the ratio of the skeletonized microtubule length to the total cell area. The final metric represents the skeletonized length of these strongly localized, high-affinity binding sites.

Microtubule-bound superTagRFP-A1aY1 sensor in the Taxol treatment experiment was quantified using Fiji Software (*Schindelin et al., 2012*). All images were sequentially processed using a 5-pixel

rolling ball radius background subtraction, 2.0-pixel Gaussian blur, and brightness and contrast set to a minimum and maximum of 50 and 990, respectively. Images were then converted to 8-bit grayscale and microtubule-bound sensor was detected using the Fiji Ridge Detection plugin (*Wagner and Hiner, 2017*) with the following settings: line width of 2.0, high contrast of 100.0, low contrast of 10.0, sigma of 1.08, lower threshold of 0.68, upper threshold of 6.97, minimum line length of 10.0, and slope method of overlap resolution. The images were binarized and total signal (IntDen) was recorded for each image and normalized to cell area.

We quantified colocalization between MAPs and Y-αCTT probes (*Figure 3*), or MAPs and detyrosination (*Figure 4*) using the threshold overlap score (TOS). This metric was calculated with the Fiji plugin EzColocalization (*Stauffer et al., 2018*). For analysis, TOS was determined based on the top 10th percentile of signal per cell, with cell boundaries defined manually.

To quantify microtubule detyrosination in cells expressing MAPs, we measured detyrosination signal exclusively on MAP-decorated microtubules, rather than on the entire microtubule network within the cells. Our rationale for this approach is that some MAPs exhibit localization only on specific subsets of microtubules. Therefore, measuring detyrosination against total cellular microtubules, as we performed previously (*Yue et al., 2023*), would inaccurately represent the net change in detyrosination directly influenced by MAP localization. To achieve this targeted measurement, we first extracted MAP-localized microtubule pixels by applying a consistent threshold across all MAP images, creating specific masks. These masks were then applied to both the total microtubule channel (visualized by anti-α-tubulin antibody DM1α) and the detyrosinated microtubule channel to obtain mean intensities for each cell. Finally, the relative intensities of detyrosination signals were plotted after normalization against the corresponding total microtubule intensities within the masked regions. In contrast, for the E254A TubA experiments, both wild-type and E254A PA-tagged TubA1A constructs showed broad incorporation into microtubules throughout the entire cell. Consequently, mean detyrosination was measured on a per-cell basis, as described previously (*Yue et al., 2023*).

## Molecular dynamics simulations

The starting point for our GDP simulations was the structure of the GDP-microtubule (PDB: 7SJ7, *LaFrance et al., 2022*). To create the GTP-microtubule, we used the GMPCPP-structure (PDB: 6DPU, *Zhang et al., 2018*) as a starting point, converted the GMPCPP to GTP, and fit the structure to the 7SJ7 microtubule so that it had the same lattice structure. C-terminal tails were added to every tubulin subunit. For all systems, rigid-body fitting was used to make a complete 13-protofilament, 3-start microtubule ring. This single ring was then shifted by the dimer repeat distance in order to make a 3-ring, 39-dimer microtubule fragment. For the GDP-microtubules, the 3-ring structure was converted to an 'infinite' microtubule (*Igaev and Grubmüller, 2020*; *Wells and Aksimentiev, 2010*) by putting it in a periodic box where the axial length was three times the dimer repeat distance. Due to the inherent twist of the GTP-microtubule, we could not create an 'infinite' microtubule since the protofilaments would not match up with their image. Instead, the 39-dimer GTP system was solvated in a box with 20 Å padding on each side.

All systems were solvated using TIP3P water and ionized with Na+ and Cl- in order to both neutralize the system and set the ionic strength to 50 mM. Simulations were carried out using NAMD (*Phillips et al., 2020*) using the CHARMM36 (*Best et al., 2012*) force field. Following minimization and heating, we performed a short 10 ns equilibrium in an NpT ensemble with 1 atm pressure at 300 K. We then ran two independent trajectories for 260 ns and 330 ns, respectively, in the GDP-microtubule system, and one 360 ns trajectory in the GTP-microtubule system, utilizing hydrogen mass repartitioning (*Hopkins et al., 2015*) to allow for 4 fs time steps. All analysis was done using bio3D (*Grant et al., 2006*) and R (https://www.r-project.org/). Images and movies were created using VMD (*Humphrey et al., 1996*).

## Data analysis, statistics, and presentation

All data were plotted and statistical tests were performed using GraphPad Prism (version 10.4.1; GraphPad Software). The statistical tests, sample size, and number of replicates used for each experiment are described in each figure legend. Figures were made in Adobe Illustrator 2025 (version 29.0; Adobe).

## Materials availability statement

All plasmids, non-commercially available reagents, and detailed protocols are available from the corresponding authors upon request. No new datasets were created.

## Acknowledgements

We thank members of the Ohi, Verhey, DeSantis, Cianfrocco, and Sept labs for discussions and advice. We thank Jakia Jannat Keya (University of Michigan) for purified KIF5C protein. We thank Jeff Moore (University of Colorado Anschutz Medical Center) for TubA1A plasmids and recombinant anti-TubB3 Tuj1 antibody.

## Additional information

### Funding

| Funder | Grant reference number | Author |
|---|---|---|
| National Institutes of Health | R35GM131744 | Kristen J Verhey |
| National Institutes of Health | R35GM153209 | Ryoma Ohi |
| National Institutes of Health | F32GM157897 | Morgan L Pimm |
| National Institutes of Health | R01GM141119 | Michael A Cianfrocco |
| National Institutes of Health | R35GM146739 | Morgan E DeSantis |
| National Institutes of Health | R01GM136822 | David Sept |
| American Cancer Society | PF-24-1320851-01-CCB | Ezekiel C Thomas |

The funders had no role in study design, data collection and interpretation, or the decision to submit the work for publication.

### Author contributions

Takashi Hotta, Morgan L Pimm, Ezekiel C Thomas, Yang Yue, Formal analysis, Validation, Investigation, Visualization, Methodology, Writing – review and editing; Patrick DeLear, Formal analysis, Validation, Investigation, Visualization, Writing – review and editing; Lynne Blasius, Validation, Investigation, Visualization, Methodology, Writing – review and editing; Michael A Cianfrocco, Morgan E DeSantis, Conceptualization, Writing – review and editing; Ryota Horiuchi, Takumi Higaki, Formal analysis, Methodology; David Sept, Conceptualization, Supervision, Funding acquisition, Project administration, Writing – review and editing; Ryoma Ohi, Kristen J Verhey, Conceptualization, Supervision, Funding acquisition, Writing – original draft, Project administration, Writing – review and editing

### Author ORCIDs

Morgan L Pimm ⓘ https://orcid.org/0000-0001-6370-1435
Ezekiel C Thomas ⓘ https://orcid.org/0000-0002-9810-0477
Patrick DeLear ⓘ https://orcid.org/0000-0002-1013-0138
Michael A Cianfrocco ⓘ https://orcid.org/0000-0002-2067-4999
Morgan E DeSantis ⓘ https://orcid.org/0000-0002-4096-8548
David Sept ⓘ https://orcid.org/0000-0003-3719-2483
Kristen J Verhey ⓘ https://orcid.org/0000-0001-9329-4981

Reviewer #1 (Public review): https://doi.org/10.7554/eLife.109308.3.sa1
Reviewer #2 (Public review): https://doi.org/10.7554/eLife.109308.3.sa2
Reviewer #3 (Public review): https://doi.org/10.7554/eLife.109308.3.sa3

Author response https://doi.org/10.7554/eLife.109308.3.sa4

## Additional files

### Supplementary files
MDAR checklist

Supplementary file 1. Sequences of oligonucleotide primers.

### Data availability
All data generated or analyzed during this study are included in the manuscript and supporting files; source data files have been provided.

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
