## [Editor Report · eLife Assessment]

This **fundamental** work reveals that the accessibility of the unstructured C-terminal tail of α-tubulin differs with the state of the microtubule lattice. Accessibility increases with the expansion of the lattice induced by GTP and certain MAPs, which can then dictate the subsequent interactions between MAPs and microtubules, and post-translational modifications of tubulin tails. The evidence supporting the conclusion is **compelling**, although the characterisation of the probes does not answer whether they directly affect the lattice or expose the C-terminal tail of α-tubulin. The probes can be used as tools in the future to study differences in microtubule lattice regulation under different conditions both in vitro and in vivo. This work will be of great interest to the cytoskeleton field.

---

## [Referee Report · Reviewer #1 (Public review)]

Summary:

This is a careful and comprehensive study demonstrating that effector-dependent conformational switching of the MT lattice from compacted to expanded deploys the alpha tubulin C-terminal tails so as to enhance their ability to bind interactors.

Strengths:

The authors use 3 different sensors for the exposure of the alpha CTTs. They show that all 3 sensors report exposure of the alpha CTTs when the lattice is expanded by GMPCPP, or KIF1C, or a hydrolysis-deficient tubulin. They demonstrate that expansion-dependent exposure of the alpha CTTs works in tissue culture cells as well as in vitro.

Appraisal:

The authors have gone to considerable lengths to test their hypothesis that microtubule expansion favours deployment of the alpha tubulin C-terminal tail, allowing its interactors, including detyrosinase enzymes, to bind. There is a real prospect that this will change thinking in the field. One very interesting possibility, touched on by the authors, is that the requirement for MAP7 to engage kinesin with the MT might include a direct effect of MAP7 on lattice expansion.

Impact:

The possibility that the interactions of MAPS and motors with a particular MT or region feed forward to determine its future interaction patterns is made much more real. Genuinely exciting.

---

## [Referee Report · Reviewer #2 (Public review)]

The unstructured α- and β-tubulin C-terminal tails (CTTs), which differ between tubulin isoforms, extend from the surface of the microtubule, are post-translationally modified, and help regulate the function of MAPs and motors. Their dynamics and extent of interactions with the microtubule lattice are not well understood. Hotta et al. explore this using a set of three distinct probes that bind to the CTTs of tyrosinated (native) α-tubulin. Under normal cellular conditions, these probes associate with microtubules only to a limited extent, but this binding can be enhanced by various manipulations thought to alter the tubulin lattice conformation (expanded or compact). These include small-molecule treatment (Taxol), changes in nucleotide state, and the binding of microtubule-associated proteins and motors. Overall, the authors conclude that microtubule lattice "expanders" promote probe binding, suggesting that the CTT is generally more accessible under these conditions. Consistent with this, detyrosination is enhanced. Mechanistically, molecular dynamics simulations indicate that the CTT may interact with the microtubule lattice at several sites, and that these interactions are affected by the tubulin nucleotide state.

Strengths and weaknesses:

Key strengths of the work include the use of three distinct probes that yield broadly consistent findings, and a wide variety of experimental manipulations (drugs, motors, MAPs) that collectively support the authors' conclusions, alongside a careful quantitative approach.

The challenges of studying the dynamics of a short, intrinsically disordered protein region within the complex environment of the cellular microtubule lattice, amid numerous other binders and regulators, should not be understated. While it is very plausible that the probes report on CTT accessibility as proposed, the possibility of confounding factors (e.g., effects on MAP or motor binding) cannot be ruled out. Sensitivity to the expression level clearly introduces additional complications. Likewise, for each individual "expander" or "compactor" manipulation, one must consider indirect consequences (e.g., masking of binding sites) in addition to direct effects on the lattice; however, this risk is mitigated by the collective observations all pointing in the same direction.

The discussion does a good job of placing the findings in context and acknowledging relevant caveats and limitations. Overall, this study introduces an interesting and provocative concept, well supported by experimental data, and provides a strong foundation for future work. This will be a valuable contribution to the field.

---

## [Referee Report · Reviewer #3 (Public review)]

Summary:

In this study, the authors investigate how the structural state of the microtubule lattice influences the accessibility of the α-tubulin C-terminal tail (CTT). By developing and applying new biosensors, they reveal that the tyrosinated CTT is largely inaccessible under normal conditions but becomes more accessible upon changes to the tubulin conformational state induced by taxol treatment, MAP expression, or GTP-hydrolysis-deficient tubulin. The combination of live imaging, biochemical assays, and simulations suggests that the lattice conformation regulates the exposure of the CTT, providing a potential mechanism for modulating interactions with microtubule-associated proteins. The work addresses a highly topical question in the microtubule field and proposes a new conceptual link between lattice spacing and tail accessibility for tubulin post-translational modification. Future work is required to distinguish CTT exposure in the microtubule lattice is sensitive to additional factors present in vivo but not in vitro.

Strengths:

(1) The study targets a highly relevant and emerging topic-the structural plasticity of the microtubule lattice and its regulatory implications.

(2) The biosensor design represents a methodological advance, enabling direct visualization of CTT accessibility in living cells.

(3) Integration of imaging, biochemical assays, and simulations provides a multi-scale perspective on lattice regulation.

(4) The conceptual framework proposed lattice conformation as a determinant of post-translational modification accessibility is novel and potentially impactful for understanding microtubule regulation.

[Editors' note: the authors have responded to the reviewers and this version was assessed by the editors.]

---

## [Author Response]

The following is the authors’ response to the original reviews

**Public Reviews:**

**Reviewer #1 (Public review):**
Summary:This is a careful and comprehensive study demonstrating that effector-dependent conformational switching of the MT lattice from compacted to expanded deploys the alpha tubulin C-terminal tails so as to enhance their ability to bind interactors.Strengths:The authors use 3 different sensors for the exposure of the alpha CTTs. They show that all 3 sensors report exposure of the alpha CTTs when the lattice is expanded by GMPCPP, or KIF1C, or a hydrolysis-deficient tubulin. They demonstrate that expansion-dependent exposure of the alpha CTTs works in tissue culture cells as well as in vitro.Weaknesses:There is no information on the status of the beta tubulin CTTs. The study is done with mixed isotype microtubules, both in cells and in vitro. It remains unclear whether all the alpha tubulins in a mixed isotype microtubule lattice behave equivalently, or whether the effect is tubulin isotype-dependent. It remains unclear whether local binding of effectors can locally expand the lattice and locally expose the alpha CTTs.Appraisal:The authors have gone to considerable lengths to test their hypothesis that microtubule expansion favours deployment of the alpha tubulin C-terminal tail, allowing its interactors, including detyrosinase enzymes, to bind. There is a real prospect that this will change thinking in the field. One very interesting possibility, touched on by the authors, is that the requirement for MAP7 to engage kinesin with the MT might include a direct effect of MAP7 on lattice expansion.Impact:The possibility that the interactions of MAPS and motors with a particular MT or region feed forward to determine its future interaction patterns is made much more real. Genuinely exciting.

We thank the reviewer for their positive response to our work. We agree that it will be important to determine if the bCTT is subject to regulation similar to the aCTT. However, this will first require the development of sensors that report on the accessibility of the bCTT, which is a significant undertaking for future work.

We also agree that it will be important to examine whether all tubulin isotypes behave equivalently in terms of exposure of the aCTT in response to conformational switching of the microtubule lattice.

We thank the reviewer for the comment about local expansion of the microtubule lattice. We believe that Figure 3 does show that local binding of effectors can locally expand the lattice and locally expose the alpha-CTTs. We have added text to clarify this.

**Reviewer #2 (Public review):**
The unstructured α- and β-tubulin C-terminal tails (CTTs), which differ between tubulin isoforms, extend from the surface of the microtubule, are post-translationally modified, and help regulate the function of MAPs and motors. Their dynamics and extent of interactions with the microtubule lattice are not well understood. Hotta et al. explore this using a set of three distinct probes that bind to the CTTs of tyrosinated (native) α-tubulin. Under normal cellular conditions, these probes associate with microtubules only to a limited extent, but this binding can be enhanced by various manipulations thought to alter the tubulin lattice conformation (expanded or compact). These include small-molecule treatment (Taxol), changes in nucleotide state, and the binding of microtubule-associated proteins and motors. Overall, the authors conclude that microtubule lattice "expanders" promote probe binding, suggesting that the CTT is generally more accessible under these conditions. Consistent with this, detyrosination is enhanced. Mechanistically, molecular dynamics simulations indicate that the CTT may interact with the microtubule lattice at several sites, and that these interactions are affected by the tubulin nucleotide state.Strengths:Key strengths of the work include the use of three distinct probes that yield broadly consistent findings, and a wide variety of experimental manipulations (drugs, motors, MAPs) that collectively support the authors' conclusions, alongside a careful quantitative approach.Weaknesses:The challenges of studying the dynamics of a short, intrinsically disordered protein region within the complex environment of the cellular microtubule lattice, amid numerous other binders and regulators, should not be understated. While it is very plausible that the probes report on CTT accessibility as proposed, the possibility of confounding factors (e.g., effects on MAP or motor binding) cannot be ruled out. Sensitivity to the expression level clearly introduces additional complications. Likewise, for each individual "expander" or "compactor" manipulation, one must consider indirect consequences (e.g., masking of binding sites) in addition to direct effects on the lattice; however, this risk is mitigated by the collective observations all pointing in the same direction.The discussion does a good job of placing the findings in context and acknowledging relevant caveats and limitations. Overall, this study introduces an interesting and provocative concept, well supported by experimental data, and provides a strong foundation for future work. This will be a valuable contribution to the field.

We thank the reviewer for their positive response to our work. We are encouraged that the reviewer feels that the Discussion section does a good job of putting the findings, challenges, and possibility of confounding factors and indirect effects in context.

**Reviewer #3 (Public review):**
Summary:In this study, the authors investigate how the structural state of the microtubule lattice influences the accessibility of the α-tubulin C-terminal tail (CTT). By developing and applying new biosensors, they reveal that the tyrosinated CTT is largely inaccessible under normal conditions but becomes more accessible upon changes to the tubulin conformational state induced by taxol treatment, MAP expression, or GTP-hydrolysis-deficient tubulin. The combination of live imaging, biochemical assays, and simulations suggests that the lattice conformation regulates the exposure of the CTT, providing a potential mechanism for modulating interactions with microtubule-associated proteins. The work addresses a highly topical question in the microtubule field and proposes a new conceptual link between lattice spacing and tail accessibility for tubulin post-translational modification.Strengths:(1) The study targets a highly relevant and emerging topic-the structural plasticity of the microtubule lattice and its regulatory implications.(2) The biosensor design represents a methodological advance, enabling direct visualization of CTT accessibility in living cells.(3) Integration of imaging, biochemical assays, and simulations provides a multi-scale perspective on lattice regulation.(4) The conceptual framework proposed lattice conformation as a determinant of post-translational modification accessibility is novel and potentially impactful for understanding microtubule regulation.Weaknesses:There are a number of weaknesses in the paper, many of which can be addressed textually. Some of the supporting evidence is preliminary and would benefit from additional experimental validation and clearer presentation before the conclusions can be considered fully supported. In particular, the authors should directly test in vitro whether Taxol addition can induce lattice exchange (see comments below).

We thank the reviewer for their positive response to our work. We have altered the text and provided additional experimental validation as requested (see below).

**Recommendations for the authors:**

**Reviewer #1 (Recommendations for the authors):**
(1) The resolution of the figures is insufficient.(2) The provision of scale bars is inconsistent and insufficient.(3) Figure 1E, the scale bar looks like an MT.(4) Figure 2C, what does the grey bar indicate?(5) Figure 2E, missing scale bar.(6) Figure 3 C, D, significance brackets misaligned.(7) Figure 3E, consider using the same alpha-beta tubulin / MT graphic as in Figure 1B.(8) Figure 5E, show cell boundaries for consistency?(9) Figure 6D, stray box above the y-axis.(11) Figure S3A, scale bar wrong unit again.(12) S3B "fixed" and mount missing scale bar in the inset.(13) S4 scale bars without scale, inconsistency in scale bars throughout all the figures.

We apologize for issues with the figures. We have corrected all of the issues indicated by the reviewer.

(10) Figure 6F, surprising that 300 mM KCL washes out rigor binding kinesin

We thank the reviewer for this important point. To address the reviewer’s concern, we have added a new supplementary figure (new Figure 6 – Figure Supplement 1) which shows that the washing step removes strongly-bound (apo) KIF5C(1-560)-Halo^554^ protein from the microtubules. In addition, we have made a correction to the Materials and Methods section noting that ATP was added in addition to the KCl in the wash buffer. We apologize for omitting this detail in the original submission. We also added text noting that the wash out step was based on Shima et al., 2018 where the observation chamber was washed with either 1 mM ATP and 300 mM K-Pipes or with 10 mM ATP and 500 mM K-Pipes buffer. In our case, the chamber was washed with 3 mM ATP and 300 mM KCl. It is likely that the addition of ATP facilitates the detachment of strongly-bound KIF5C.

(14) Supplementary movie, please identify alpha and beta tubules for clarity. Please identify residues lighting up in interaction sites 1,2 & 3.

Thank you for the suggestions. We have made the requested changes to the movie.

**Reviewer #2 (Recommendations for the authors):**
There appear to have been some minor issues (perhaps with .pdf conversion) that leave some text and images pixelated in the .pdf provided, alongside some slightly jarring text and image positioning (e.g., Figure 5E panels). The authors should carefully look at the figures to ensure that they are presented in the clearest way possible.

We apologize for these issues with the figures. We have reviewed the figures carefully to ensure that they are presented in the clearest way possible.

The authors might consider providing a more definitive structural description of compact vs expanded lattice, highlighting what specific parameters are generally thought to change and by what magnitude. Do these differ between taxol-mediated expansion or the effects of MAPs?

Thank you for the suggestion. We have added additional information to the Introduction section.

**Reviewer #3 (Recommendations for the authors):**
(1) Figure 1 should include a schematic overview of all constructs used in the study. A clear illustration showing the probe design, including the origin and function of each component (e.g., tags, domains), would improve clarity.

Thank you for the suggestion. We have added new illustrations to Figure 1 showing the origin and design (including domains and tags) of each probe.

(2) Add Western blot data for the 4×CAP-Gly construct to Figure 1C for completeness.

We thank the reviewer for this suggestion. We carried out a far-western blot using the purified 4xCAPGly-mEGFP protein to probe GST-Y, GST-DY, and GST-DC2 proteins (new Figure 1 – Figure Supplement 1C). We note that some bleed-through signal can be seen in the lanes containing GST-ΔY and GST-ΔC2 protein due to the imaging requirements and exposure needed to visualize the 4xCAPGly-mEGFP protein. Nevertheless, the blot shows that the purified CAPGly sensor specifically recognizes the native (tyrosinated) CTT sequence of TUBA1A.

(3) Essential background information on the CAP-Gly domain, SXIP motif, and EB proteins is missing from the Introduction. These concepts appear abruptly in the Results and should be properly introduced.

Thank you for the suggestion. We have added additional information to the Introduction section about the CAP-Gly domain. However, we feel that introducing the SXIP motif and EB proteins at this point would detract from the flow of the Introduction and we have elected to retain this information in the Results section when we detail development of the 4xCAPGly probe.

(4) In Figure 2E, it remains possible that the CAP-Gly domain displacement simply follows the displacement of EB proteins. An experiment comparing EB protein localization upon Taxol treatment would clarify this relationship.

We thank the reviewer for raising this important point. To address the reviewer’s concern, we utilized HeLa cells stably expressing EB3-GFP. We performed live-cell imaging before and after Taxol addition (new Figure 2 – Figure Supplement 1C). EB3-EGFP was lost from the microtubule plus ends within minutes and did not localize to the now-expanded lattice.

(5) Statements such as "significantly increased" (e.g., line 195) should be replaced with quantitative information (e.g., "1.5-fold increase").

We have made the suggested changes to the text.

(6) Phrases like "became accessible" should be revised to "became more accessible," as the observed changes are relative, not absolute. The current wording implies a binary shift, whereas the data show a modest (~1.5-fold) increase.

We have made the suggested changes to the text.

(7) Similarly, at line 209, the terms "minimally accessible" versus "accessible" should be rephrased to reflect the small relative change observed; saturation of accessibility is not demonstrated.

We have made the suggested changes to the text.

(8) Statements that MAP7 "expands the lattice" (line 222) should be made cautiously; to my knowledge, that has not been clearly established in the literature.

We thank the reviewer for this important comment. We have added text indicating that MAP7’s ability to induce or presence an expanded lattice has not been clearly established.

(9) In Figures 3 and 4, the overexpression of MAP7 results in a strikingly peripheral microtubule network. Why is there this unusual morphology?

The reviewer raises an interesting question. We are not sure why the overexpression of MAP7 results in a strikingly peripheral microtubule network but we suspect this is unique to the HeLa cells we are using. We have observed a more uniform MAP7 localization in other cell types [e.g. COS-7 cells (Tymanskyj et al. 2018)], consistent with the literature [e.g. BEAS-2B cells (Shen and Ori-McKenney 2024), HeLa cells (Hooikaas et al. 2019)].

(10) In Supplementary Figure 5C, the Western blot of detyrosination levels is inconsistent with the text. Untreated cells appear to have higher detyrosination than both wild-type and E254A-overexpressing cells. Do you have any explanation?

We thank the reviewer for this important comment. We do not have an explanation at this point but plan to revisit this experiment. Unfortunately, the authors who carried out this work recently moved to a new institution and it will be several months before they are able to get the cell lines going and repeat the experiment. We thus elected to remove what was Supp Fig 5C until we can revisit the results. We believe that the important results are in what is now Figure 5 - Figure Supplement 1A,B which shows that the expression levels of the WT and E254E proteins are similar to each other.

(11) The image analysis method in Figures 5B and 5D requires clarification. It appears that "density" was calculated from skeletonized probe length over total area, potentially using a strict intensity threshold. It looks like low-intensity binding has been excluded; otherwise, the density would be the same from the images. If so, this should be stated explicitly. A more appropriate analysis might skeletonize and integrate total fluorescence intensity relative to the overall microtubule network.

We have added additional information to the Materials and Methods section to clarify the image analysis. We appreciate the reviewer’s valuable feedback and the suggestion to use the integrated total fluorescence intensity, which is a theoretically sound approach. While we agree that integrated intensity is a valid metric for specific applications, its appropriate use depends on two main preconditions:

(1) Consistent microscopy image acquisition conditions.

(2) Consistent probe expression levels across all cells and experiments.

We successfully maintained consistent image acquisition conditions (e.g., exposure time) throughout the experiment. However, despite generating a stably-expressing sensor cell lines to minimize variation, there remains an inherent, biological variability in probe expression levels between individual cells. Integrated intensity is highly susceptible to this cell-to-cell variability. Relying on it would lead to a systematic error where differences in the total amount of expressed probe would be mistaken for differences in Y-aCTT accessibility.

The density metric (skeletonized probe length / total cell area) was deliberately chosen as it serves as a geometric measure rather than an intensity-based normalization. The density metric quantifies the *proportion* of the microtubule network that is occupied by Y-aCTT-labeled structures, independent of fluorescence intensity. Thus, the density metric provides a more robust and interpretable measure of Y-aCTT accessibility under the variable expression conditions inherent to our experimental system. Therefore, we believe that this geometric approach represents the most appropriate analysis for our image dataset.

(12) In Figure 5D, the fold-change data are difficult to interpret due to the compressed scale. Replotting is recommended. The text should also discuss the relative fold changes between E254A and Taxol conditions, Figure 2H.

We appreciate the reviewer's insightful comment. We agree that the presence of significant outliers led to a compressed Y-axis scale in Figure 5D, obscuring the clear difference between the WT-tubulin and E254A-tubulin groups. As suggested, we have replotted Figure 5D using a broken Y-axis to effectively expand the relevant lower range of the data while still accurately representing all data points, including the outliers. We believe that the revised graph significantly enhances the clarity and interpretability of these results. For Figure 2, we have added the relative fold changes to the text as requested.

(13) Figure 6. The authors should directly test in vitro whether Taxol addition can induce lattice exchange, for example, by adding Taxol to GDP-microtubules and monitoring probe binding. Including such an assay would provide critical mechanistic evidence and substantially strengthen the conclusions. I was waiting for this experiment since Figure 2.

We thank the reviewer for this suggestion. As suggested, we generated GDP-MTs from HeLa tubulin and added it to two flow chambers. We then flowed in the YL1/2^Fab^-EGFP probe into the chambers in the presence of DMSO (vehicle control) or Taxol. Static images were taken and the fluorescence intensity of the probe on microtubules in each chamber was quantified. There was a slight but not statistically significant difference in probe binding between control and Taxol-treated GDP-MTs (Author response image 1). While disappointing, these results underscore our conclusion (Discussion section) that microtubule assembly in vitro may not produce a lattice state resembling that in cells, either due to differences in protofilament number and/or buffer conditions and/or the lack of MAPs during polymerization.

**Author response image 1. sa4fig1:** 

References

Hooikaas, P. J., Martin, M., Muhlethaler, T., Kuijntjes, G. J., Peeters, C. A. E., Katrukha, E. A., Ferrari, L., Stucchi, R., Verhagen, D. G. F., van Riel, W. E., Grigoriev, I., Altelaar, A. F. M., Hoogenraad, C. C., Rudiger, S. G. D., Steinmetz, M. O., Kapitein, L. C. and Akhmanova, A. (2019). MAP7 family proteins regulate kinesin-1 recruitment and activation. J Cell Biol, 218, 1298-1318.

Shen, Y. and Ori-McKenney, K. M. (2024). Microtubule-associated protein MAP7 promotes tubulin posttranslational modifications and cargo transport to enable osmotic adaptation. Dev Cell, 59, 1553-1570.

Tymanskyj, S. R., Yang, B. H., Verhey, K. J. and Ma, L. (2018). MAP7 regulates axon morphogenesis by recruiting kinesin-1 to microtubules and modulating organelle transport. Elife, 7.